# A neuroepithelial wave of BMP signalling drives anteroposterior specification of the tuberal hypothalamus

Kavitha Chinnaiya[1], Sarah Burbridge[1], Aragorn Jones[1], Dong Won Kim[2], Elsie Place[1], Elizabeth Manning[1], Ian Groves[1], Changyu Sun[2], Matthew Towers[1], Seth Blackshaw[2,3,4,5,6,7], Marysia Placzek[1,8,9]*

[1]School of Biosciences, University of Sheffield, Sheffield, United Kingdom; [2]Solomon H. Snyder Department of Neuroscience, Johns Hopkins University School of Medicine, Baltimore, United States; [3]Department of Psychiatry and Behavioral Science, Johns Hopkins University School of Medicine, Baltimore, United States; [4]Department of Ophthalmology, Johns Hopkins University School of Medicine, Baltimore, United States; [5]Department of Neurology, Johns Hopkins University School of Medicine, Baltimore, United States; [6]Institute for Cell Engineering, Johns Hopkins University School of Medicine, Baltimore, United States; [7]Kavli Neuroscience Discovery Institute, Johns Hopkins University School of Medicine, Baltimore, United States; [8]Bateson Centre, University of Sheffield, Sheffield, United Kingdom; [9]Neuroscience Institute, University of Sheffield, Sheffield, United Kingdom

*For correspondence:
m.placzek@sheffield.ac.uk

Competing interest: The authors declare that no competing interests exist.

**Abstract** The tuberal hypothalamus controls life-supporting homeostatic processes, but despite its fundamental role, the cells and signalling pathways that specify this unique region of the central nervous system in embryogenesis are poorly characterised. Here, we combine experimental and bioinformatic approaches in the embryonic chick to show that the tuberal hypothalamus is progressively generated from hypothalamic floor plate-like cells. Fate-mapping studies show that a stream of tuberal progenitors develops in the anterior-ventral neural tube as a wave of neuroepithelial-derived BMP signalling sweeps from anterior to posterior through the hypothalamic floor plate. As later-specified posterior tuberal progenitors are generated, early specified anterior tuberal progenitors become progressively more distant from these BMP signals and differentiate into tuberal neurogenic cells. Gain- and loss-of-function experiments in vivo and ex vivo show that BMP signalling initiates tuberal progenitor specification, but must be eliminated for these to progress to anterior neurogenic progenitors. scRNA-Seq profiling shows that tuberal progenitors that are specified after the major period of anterior tuberal specification begin to upregulate genes that characterise radial glial cells. This study provides an integrated account of the development of the tuberal hypothalamus.

## Editor's evaluation

The manuscript provides a comprehensive insight into the development of the tuberal hypothalamus of the chick by carefully analyzing the expression patterns of a plethora of proteins involved and perturbation of BMP signaling. The fundamental findings presented here substantially advance our understanding of the development of the chick tuberal hypothalamus from floor plate- like cells, mediated by an anterior to posterior wave of neuroepitelial-derived BMP signaling. Using bioinformatical and in situ profiling, fate mapping and tissue explants the authors present compelling evidence supporting their conclusion.

## Introduction

The tuberal hypothalamus constitutes the antero-ventral hypothalamus. Like the rest of the hypothalamus, it is an evolutionarily ancient part of the forebrain. Its core neurons and neurohormones have been conserved due to their roles in the central regulation of behaviours and homeostatic physiological processes that are essential to life (*Saper and Lowell, 2014*). Key behaviours and processes regulated by tuberal hypothalamic neurons include energy balance, growth, stress, reproduction, and both defensive and offensive aggression. In addition to harbouring neurons, the tuberal hypothalamus houses radial glial-like tanycytes, which regulate the dialogue between the brain and the body, and are central players in the control of energy metabolism and reproduction (*Bolborea and Langlet, 2021*; *Rohrbach et al., 2021*; *Yoo et al., 2020*). Tanycytes also function as stem and progenitor cells, and their position defines an adult hypothalamic stem cell niche (*Haan et al., 2013*; *Lee et al., 2012*; *Robins et al., 2013*; *Yoo et al., 2021*; *Yoo and Blackshaw, 2018*).

In recent years, a combination of scRNA-Seq analyses and large-scale expression profiling has enabled the characterisation of major tuberal hypothalamic neuronal and tanycyte subtypes (*Campbell et al., 2017*; *Chen et al., 2017*; *Kim et al., 2020*; *Romanov et al., 2020*; *Shimogori et al., 2010*; *Sullivan et al., 2022*). scRNA-Seq studies have, additionally, identified markers of the major tuberal hypothalamic progenitor subtypes, and in combination with classic genetic studies, have provided insights into some of the key molecular regulators of tuberal hypothalamic neuronal and tanycyte specification (*Bedont et al., 2014*; *Blackshaw et al., 2010*; *Kim et al., 2022*; *Kim et al., 2020*; *Lee et al., 2018*; *Ma et al., 2021*; *Miranda-Angulo et al., 2014*; *Newman et al., 2018a*; *Newman et al., 2018b*; *Romanov et al., 2020*; *Salvatierra et al., 2014*). However, despite these advances, we do not have an integrated understanding of tuberal development. Efforts have been hampered by the small size of the nascent hypothalamus, its complex morphology and its early development, in comparison to other well-characterised regions of the central nervous system (CNS). Further, while studies have suggested how tuberal radial glial cells give rise to tanycytes (*Haan et al., 2013*; *Lee et al., 2012*; *Robins et al., 2013*; *Yoo et al., 2021*), the lineage relationship of tuberal radial glia and other sets of tuberal progenitors is not known.

A recent comprehensive scRNA Seq and profiling analysis in the chick, conducted over Hamburger-Hamilton (HH) stages 8–21, revealed that tuberal progenitors are first detected at HH10 and tuberal neurons at HH13 (*Kim et al., 2022*). Following the onset of neurogenesis, two tuberal domains can be distinguished: an anterior neurogenic domain (*SIX6/SHH/POMC*) and a posterior domain (*TBX2/FGF10/RAX*) that overlies Rathke's pouch (*Kim et al., 2022*; *Manning et al., 2006*). We do not understand how these domains develop. However, interrogation of the aggregated scRNA-Seq data sets, using RNA velocity, revealed a distinctive differentiation trajectory for tuberal hypothalamic neurogenesis that begins in *RAX/FGF10* progenitor cells, then passes through *SIX6/ASCL1* neurogenic precursors. Further, fate-mapping studies suggest that cells in the anterior neurogenic domain originate from cells that lie more posteriorly within the neuraxis (*Fu et al., 2017*). While these studies suggest that anterior neurogenic precursors derive from progenitors in the posterior domain, key questions remain. Do anterior and posterior domain cells originate from a long-term self-renewing multipotent stem-like pre-tuberal population, or from a series of transitory pre-tuberal progenitor cells, each with limited potential? And what are the molecular characteristics of pre-tuberal progenitor population(s)? New insights show that tuberal cells are first detected immediately anterior to hypothalamic floor plate-like (HypFP) cells (*Kim et al., 2022*) previously termed rostral diencephalic ventral midline cells (*Dale et al., 1997*; *Dale et al., 1999*), but the lineage relationship of these two cell groups has not been examined.

Not only is the ontogeny of tuberal hypothalamic cells unclear, but so, too are the signalling pathways that direct their specification. Pharmacological studies in the chick indicate that BMPs and low WNT levels direct the specification of posterior domain cells (*Manning et al., 2006*), whereas SHH and Notch signalling specify anterior neurogenic domain cells and neurogenesis (*Dupé et al., 2011*; *Fu et al., 2017*; *Ratié et al., 2014*, *Ratié et al., 2013*; *Ware et al., 2016*). These studies describe an intimate interaction between BMP signalling and *SHH* expression, demonstrating that BMPs downregulate *SHH* in posterior domain progenitors by inducing the transcriptional repressor *TBX2* (*Manning et al., 2006*). But overall, we do not understand the role of BMP signalling in tuberal development. No study has systematically investigated active BMP signalling in the context of the developing anterior and posterior neural progenitor tuberal domains. Studies in mouse and zebrafish suggest that

the key domains, transcription factors and signalling molecules that characterise the chick tuberal hypothalamus are conserved, confirm the importance of SHH signalling in the specification of anterior neurogenic domain cells and neurogenesis, confirm the regulation of *SHH* by TBX2/3, and indicate that a delicate balance between SHH and BMP is needed for tuberal hypothalamic development (*Blaess et al., 2014*; *Brinkmeier et al., 2007*; *Corman et al., 2018*; *Goodman et al., 2020*; *Jeong et al., 2008*; *Kim et al., 2020*; *Muthu et al., 2016*; *Newman et al., 2018a*; *Newman et al., 2018b*; *Orquera et al., 2016*; *Pontecorvi et al., 2008*; *Ratié et al., 2013*; *Shimogori et al., 2010*; *Szabó et al., 2009*). However, as in the chick, these studies have not provided an integrated understanding of tuberal hypothalamic development.

Here, we examine the spatio-temporal development of the tuberal hypothalamus. We show that a stream of tuberal progenitors is generated over a sustained period as a wave of canonical BMP signalling, sustained by neuroepithelial-intrinsic BMPs, sweeps from anterior to posterior through HypFP cells. As early specified progenitors become anteriorly distanced from the source of the BMP signals, they lose pSMAD1/5/8, downregulate posterior domain markers, and upregulate anterior domain markers that include *SHH* and neurogenic markers. Early born posterior progenitors are therefore converted to anterior neurogenic progenitors. The tuberal hypothalamus is therefore generated from a series of transitory progenitors, characterised first by expression of floor plate, then posterior tuberal and finally anterior tuberal markers. Their emergence and growth result in a steady increase in distance between cell populations expressing ventral telencephalic and floor plate markers. Our work suggests that posterior progenitors continue to be generated beyond the major period of anterior tuberal specification, and scRNA-Seq profiling reveals that over time, the transcriptional profile of posterior tuberal progenitors changes. We describe a gradual upregulation of candidate molecular regulators of a gliogenic programme that likely initiates the transformation of posterior tuberal progenitors to the stem-like radial glial cells that characterise the late-embryonic tuberal hypothalamus.

## Results

### A spatio-temporal pattern of tuberal neurogenesis

Our previous scRNA-Seq study indicated that the chick tuberal hypothalamus develops and explodes rapidly over HH10-HH13; RNA velocity, further, revealed a distinctive differentiation trajectory for tuberal hypothalamic neurogenesis, passing from *SIX6/FGF10*[+ve] progenitor cells, via *SIX6/ASCL1/ISL1*[+ve] neurogenic precursors, to terminate in *SIX6/NR5A1/POMC* [+ve] neurons (*Kim et al., 2022*). To examine the spatio-temporal development of the chick tuberal hypothalamus, we therefore profiled tuberal markers over time and relative to markers of surrounding brain regions, through multiplex hybridisation chain reaction (HCR) in situ analyses. The tuberal marker, *SIX6*, is barely detected at HH8 (*Figure 1A*; *Figure 1—figure supplement 1D*), and instead, HypFP cells, defined through *FOXA2*[(low)]/*NKX2-1/SHH* expression (*Kim et al., 2022*) extend to almost the tip of the neuraxis at this stage (*Figure 1A and F*; white arrows). By HH10, *SIX6* is detected within anterior-most hypothalamic floor plate-like cells (aHypFP cells), in a flat portion of anterior ventral neuroepithelium (*Figure 1B, G, H, I*, yellow arrows) that lies between *FOXG1*[+ve] telencephalic cells and a characteristic flexure composed of more posterior HypFP cells (*Figure 1B, H, I*, blue arrows). Over HH10-HH20, the *SIX6* domain lengthens and *SIX6* becomes graded, high expression marking the anterior tuberal domain and weak/no expression in the posterior tuberal domain that overlies Rathke's pouch (*Figure 1C–E*).

We next analysed the expression of *SIX6* relative to *TBX2*, a marker of the future posterior tuberal domain (*Kim et al., 2022*; *Manning et al., 2006*). We aimed to precisely pinpoint the time at which both genes are first detected, and determine when discrete anterior (*SIX6*[+ve]) and posterior (*TBX2*[+ve]) domains are detected. *SIX6* and *TBX2* were analysed simultaneously with *SHH*, which marks HypFP cells, underlying prechordal mesoderm, and later on, anterior neurogenic cells. Due to the complex morphology that we had noted, even at early stages, we analysed expression patterns in transverse and sagittal sections, and in wholemount neural tubes and hemi-views (*Figure 1—figure supplement 1A–C*).

*TBX2* and *SIX6* are first detected between HH8–HH9, and, surprisingly, show almost identical expression in aHypFP cells (*Figure 1J, K, O and P*; *Figure 1—figure supplement 1D–E''*). Thereafter, *SIX6* and *TBX2* gradually resolve to mark distinct anterior and posterior domains (*Figure 1L–N, Q and R*; *Figure 1—figure supplement 1F–I*). By HH13, the *SIX6*[+ve] domain has expanded considerably,

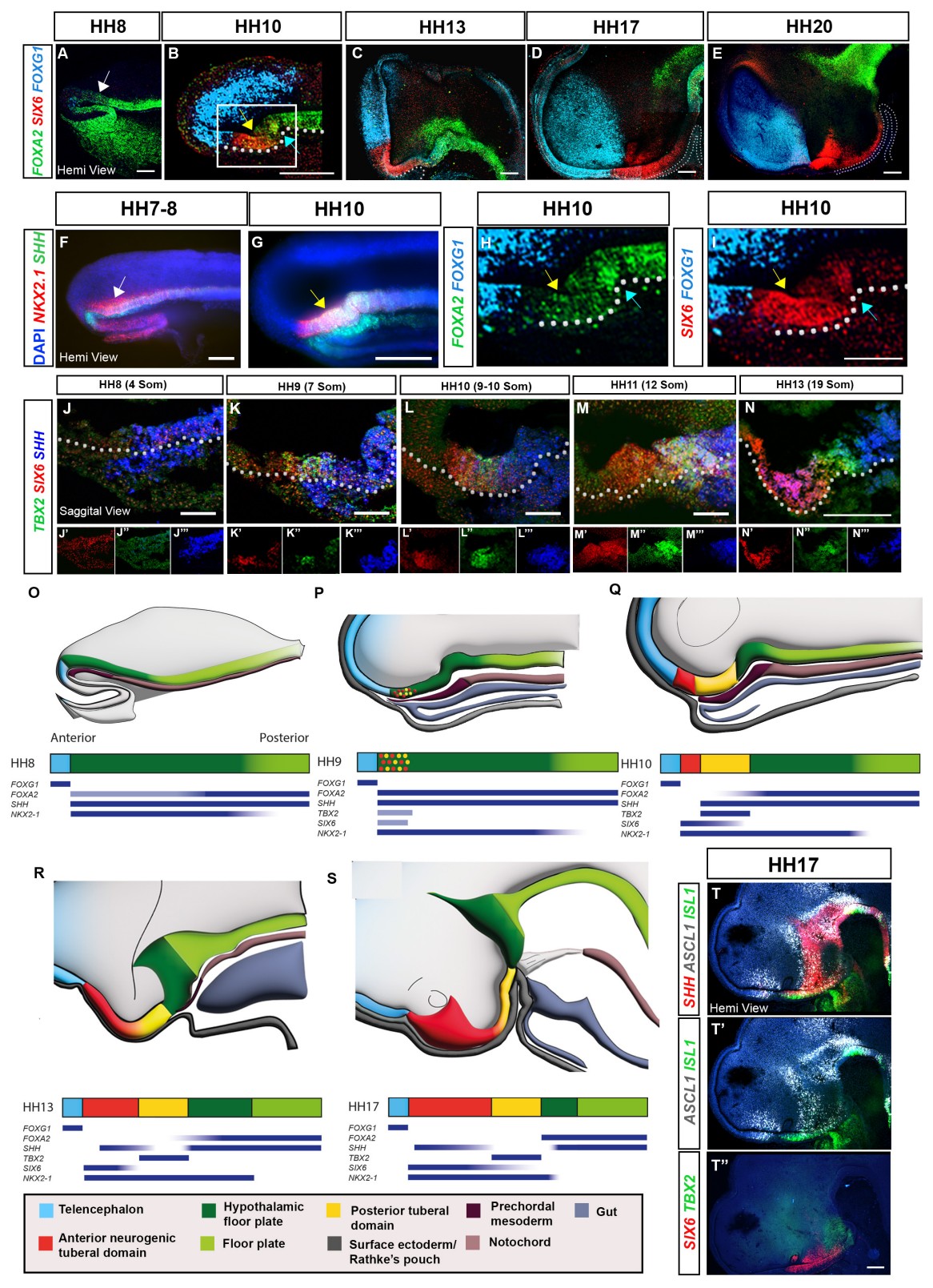

**Figure 1.** Spatio-temporal development of the tuberal hypothalamus. (**A–I**) Maximum intensity projections of hemi-dissected HH8–HH20 heads (hemi-views) after HCR for *FOXA2/SIX6/FOXG1* or *SHH/NKX2-1* (white arrows: aHypFP cells; yellow arrows: tuberal markers within aHypFP cells; blue arrows: HH10 flexure). (**H, I**) Double-channel views of boxed region in (**B**). (**J–N**) Sagittal sections of HH8–HH13 embryos after HCR for *TBX2/SIX6/SHH*. Individual channels are shown in (**J'–N'''**). (**O–S**) Schematics show hemi-views (top row) or bar representation (second row) of neuroectoderm gene

*Figure 1 continued*

expression domains in HH8–HH17 embryos; colours represent anterior neurogenic (red) and posterior (yellow) tuberal hypothalamic progenitor domains, relative to telencephalon (blue), HypFP (dark green), floor plate (green), oral ectoderm/Rathke's pouch (dark grey), anterior (gut) endoderm (grey), prechordal mesoderm (dark brown) and notochord (brown). Blue bars beneath schematics show selected gene expression profiles. (**T, T″**) Hemi-views of HH17 head after HCR for *SHH/ASCL1/ISL* (**T**), re-probed in a second round with *SIX6/TBX2* (**T″**). White dots in (C-E) outline Rathke's pouch, and white dots in (B and I-N) outline basal neural tube. Scale bars: 100 μm. Each panel shows a representative image from n=3–5 embryos. HCR, hybridisation chain reaction.

The online version of this article includes the following figure supplement(s) for figure 1:

**Figure supplement 1.** Profiling developing tuberal progenitors.

and is composed of *SIX6*$^{+ve}$ and *SIX6/SHH*$^{+ve}$ cells, while *SHH* starts to be downregulated in *TBX2*$^{+ve}$ posterior cells (***Figure 1N and R***). By HH17, the *SIX6/SHH* domain is neurogenic (*ASCL1*$^{+ve}$*)*, and anterior *SIX6*$^{+ve}$ cells are more differentiated, expressing the neuronal precursor cell marker, *ISL1* (***Figure 1T–T″***).

In summary, tuberal markers that characterise the future anterior (*SIX6*$^{+ve}$) and posterior (*TBX2*$^{+ve}$) tuberal domains are first detected at HH8-9, and are expressed together within aHypFP cells (***Figure 1O and P***). Almost immediately, the *SIX6*$^{+ve}$ and *TBX2*$^{+ve}$ expression domains begin to resolve, and the *SIX6*$^{+ve}$ domain lengthens, resulting in a marked separation between telencephalic and floor plate markers (***Figure 1Q-S***). In this way, discrete tuberal domains become apparent from anterior to posterior: *SIX6/ISL1*$^{+ve}$ and *SIX6/SHH/ASCL1*$^{+ve}$ neurogenic progenitors, *TBX2*$^{+ve}$ tuberal progenitors, and *FOXA2/SHH* $^{+ve}$ HypFP cells. Taken together, this implies that the differentiation trajectory of tuberal neurogenesis predicted using RNA velocity studies can be mapped out in vivo, with the early/least differentiated parts of the trajectory mapping to more posterior locations and more differentiated cells found more anteriorly.

## An anterior-posterior wave of BMP signalling passes through HypFP cells

*TBX2* is upregulated by BMPs in the developing tuberal hypothalamus (***Manning et al., 2006***) prompting us to investigate the activity (phosphorylation) of BMP effector SMADs 1, 5 and 8. pSMAD1/5/8 is detected immediately after HH8 in aHypFP cells (***Figure 2A–B‴***; ***Figure 2—figure supplement 1A***). Thereafter, and at least until HH17 (the latest stage analysed) pSMAD1/5/8 is always detected in aHypFP cells and cells just anterior to these (***Figure 2C–C″″***; ***Figure 2—figure supplement 1B***). Analysis of pSMAD1/5/8 in combination with a range of markers shows that pSMAD1/5/8$^{+ve}$ cells overlie Rathke's pouch and/or its *LHX3*$^{+ve}$ surface ectoderm precursors (***Figure 2D–F***, yellow dashed line), are distinct from *FOXG1*$^{+ve}$ telencephalic progenitors, are posterior to *ASCL1*$^{+ve}$ anterior tuberal neurogenic cells markers, and overlap with, but extend posterior to *TBX2*$^{+ve}$ cells (***Figure 2—figure supplement 1C–E***). In summary, pSMAD1/5/8 is first detected just after HH8, and at all times thereafter straddles aHypFP cells and posterior-most cells of the lengthening tuberal hypothalamus.

The profile of pSMAD1/5/8 suggests an association with a tuberal progenitor state, and suggests two alternate hypotheses for the generation of the tuberal hypothalamus. First, pSMAD1/5/8 could mark a stable population of self-renewing cells, which are displaced posteriorly as other tuberal progenitors are generated from, and anterior to them (as suggested in ***Fu et al., 2017***). Alternatively, pSMAD1/5/8 activity may travel through the neuroepithelium in an anterior to posterior wave, leaving tuberal cells in its wake.

To distinguish between these possibilities, we performed fate-mapping experiments. The distinct morphology of the HH10 neuroepithelium allowed us to accurately target pSMAD1/5/8$^{+ve}$ aHypFP cells or more posterior HypFP cells. In some cases, we were able to simultaneously target aHypFP cells and underlying ectoderm cells (***Figure 2—figure supplement 2A–D***). When DiI was simultaneously targeted to aHypFP cells and underlying ectoderm at HH10, examination of embryos at HH17 showed that tissues that were adjacent at HH10 had moved out of register, the neuroectoderm displaced relatively forward, and the surface ectoderm-derived Rathke's pouch relatively backwards (***Figure 2G–J***). Further processing by HCR in situ hybridisation showed that cells within the pSMAD1/5/8$^{+ve}$ aHypFP region at HH10 gave rise to *SHH* $^{+ve}$ anterior progenitors at HH17 (***Figure 2K and K'***). Additional analyses showed that cells within the aHypFP region at HH10 gave rise to regions located anterior to *TBX2*$^{+ve}$ progenitors (***Figure 2—figure supplement 2E, F***). This implies that anterior neurogenic progenitors

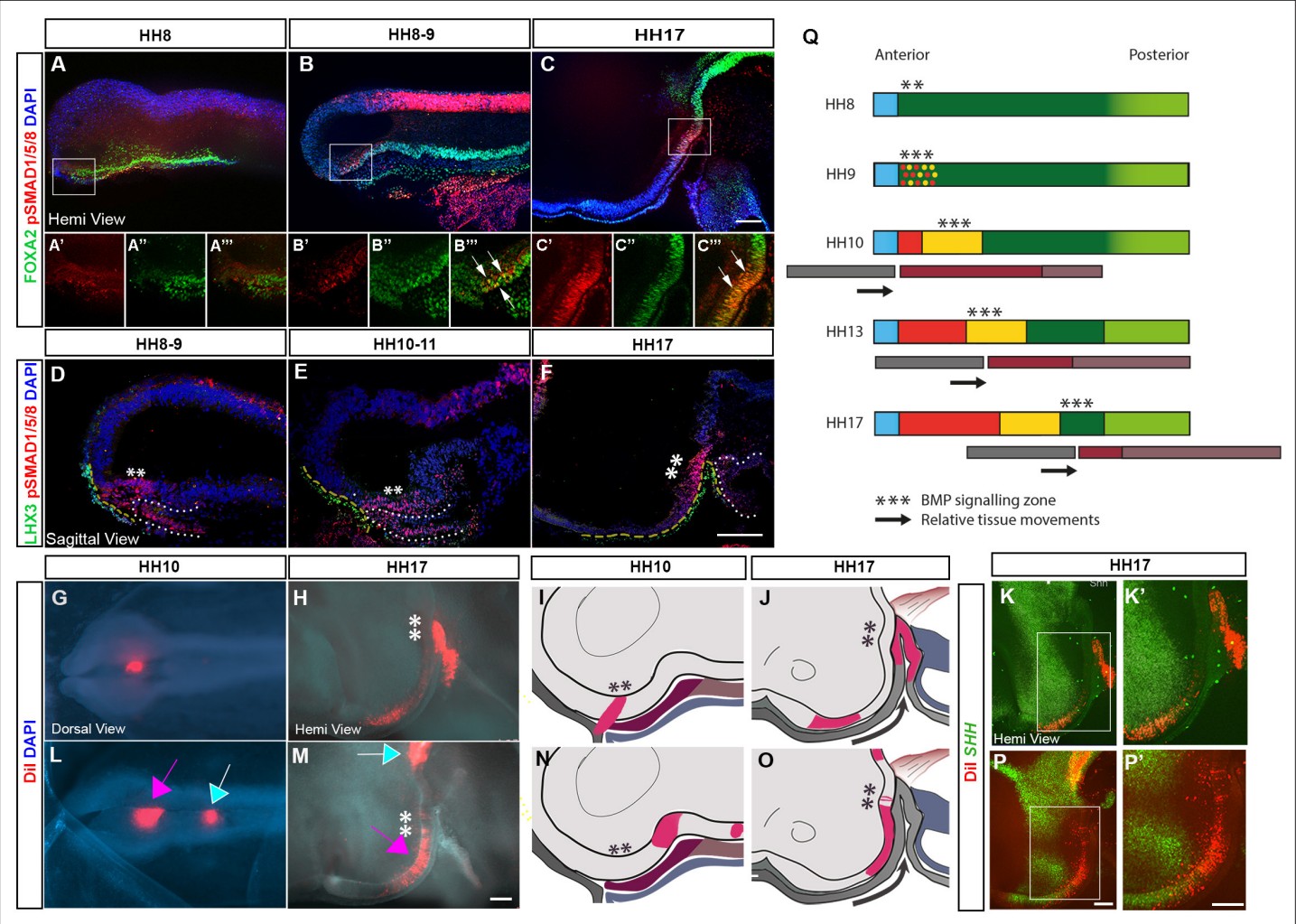

**Figure 2.** Tuberal progenitors originate from HypFP cells in a conveyor-belt manner. (**A–F**) Hemi-views (**A–C**) and sagittal sections (**D–F**) from HH8 to HH17 embryos, immunolabelled to detect pSMAD1/5/8 and FOXA2 (**A–C**): single channel views of boxed regions shown in (**A'–C'''**) or pSMAD1/5/8 and LHX3 (**D–F**). Yellow dashes and white dots outline the surface ectoderm/Rathke's pouch and anterior gut, respectively. (**G, I**) Dorsal view (**G**) of an HH10 embryo after targeting DiI to aHypFP cells and underlying ectoderm (shown schematically in side view in (**I**)). (**H**) Hemi-view of the same embryo at HH17, schematised in (**J**). (**K, K'**) Hemi-view of the same embryo, after HCR in situ hybridisation. DiI is located in *SHH* +ve anterior neurogenic cells. (**L, N**) Dorsal view (**L**) of an HH10 embryo, after targeting DiI to posterior HypFP cells (pink arrow), and cells at the border of HypFP and floor plate (blue arrow) (shown schematically in side view in (**N**)). (**M–P'**) Hemi-views of the same embryo at HH17, visualised for DiI (**M**), shown schematically in (**O**) and after HCR in situ hybridisation (**P, P'**). DiI is located in *SHH*-ve tuberal cells, petering out in *SHH*+ve anterior neurogenic cells. Asterisks ** indicate the location of pSMAD1/5/8 cells. Black arrows in (**J, O**) show relative posterior displacement of pSMAD1/5/8 cells and Rathke's pouch compared to neuroectoderm. (**Q**) A model for the generation of tuberal hypothalamus. A wave of BMP signalling (***) tracks posteriorly through HypFP cells, generating tuberal progenitors in its wake. Black arrows show relative posterior movement of underlying tissues Rathke's pouch (grey bars) in comparison to the neuroectoderm (thicker coloured bars). Colours as in *Figure 1O-S*. All scale bars = 100 µm. Each panel shows a representative image from n=4 embryos/stage for immunolabelling, and from n > 15 embryos for fate-mapping. HCR, hybridisation chain reaction.

The online version of this article includes the following figure supplement(s) for figure 2:

**Figure supplement 1.** pSMAD1/5/8+ve cells mark the posterior-most tuberal domain.

**Figure supplement 2.** Fate-mapping identifies the origin of tuberal hypothalamic cells.

are derived from progenitors that transiently activate BMP signalling, and that pSMAD1/5/8 does not mark a stable (label-retaining) population. In turn, this suggests that pSMAD1/5/8 must instead mark successively more posterior HypFP cells.

To confirm this, we targeted more posterior HypFP cells (*Figure 2L, N*, pink arrow; *Figure 2— figure supplement 2G*) that are pSMAD1/5/8 negative at HH10 (*Figure 2—figure supplement 1*). We found that these give rise to posterior tuberal progenitors at HH17 (*SIX6*/SHH-ve, *FGF10*+ve)

(*Figure 2M, O, P, P'*, pink arrow; *Figure 2—figure supplement 2H*), sometimes even contributing to (posterior-most) anterior tuberal progenitors (*SIX6/SHH*+ve, *FGF10*-ve) (*Figure 2M, O, P and P'*; *Figure 2—figure supplement 2H*). Cells that lie even more posteriorly at HH10 fate-mapped to the diencephalic floor plate (*Figure 2L and M*, blue arrow). Double- or triple-label analyses with DiI and DiO, further, suggested that cells maintain their relative antero-posterior spatial positions within the neuroectoderm between HH10 and HH17, with no evidence of mixing (*Figure 2—figure supplement 2I–N*), an idea supported through time-lapse imaging of DiI/DiO-targeted isolated neural tubes (*Figure 2—figure supplement 2O–R*).

These results suggest that the tuberal hypothalamus is generated through a conveyor-belt mechanism, through a programme that is initiated when canonical BMP signalling is activated in aHypFP cells, and then relayed back through more posterior HypFP cells (schematised in *Figure 2Q*). The appearance of pSMAD1/5/8+ve nuclei is associated with the upregulation of tuberal progenitor markers and the loss of *FOXA2*. As tuberal cells are generated, the zone of active BMP signalling, and underlying tissues, move relatively posteriorly. Newly specified posterior tuberal markers remain closely associated with the pSMAD1/5/8+ve zone, while the anterior tuberal domain expands. We conclude that tuberal progenitor cells are generated sequentially, with prospective anterior tuberal cells moving through the BMP signalling zone earlier than prospective posterior tuberal cells.

## A neuroepithelial-intrinsic programme drives tuberal hypothalamic development from HH10 onward

From HH9 to HH17, *LHX3*+ve ectoderm/Rathke's pouch tip cells express *BMP2* and *BMP7* (*Figure 3A–F*, blue arrows). This raises the possibility that Rathke's pouch elicits BMP signalling in HypFP cells, the posterior-directed growth of Rathke's pouch driving the anterior-to-posterior wave of transient pSMAD1/5/8. However, *BMP2* and *BMP7* are also expressed in hypothalamic cells (*Figure 3A–F*, pink arrows; *Kim et al., 2022*). Indeed, HCR analysis shows that *TBX2* is first upregulated in cells that express high levels of *BMP2* and lie just anterior to *BMP7*+ve cells (*Figure 3—figure supplement 1A–C*). Neuroepithelial-intrinsic BMPs could therefore drive tuberal development independently of signals from Rathke's pouch and/or other extrinsic tissues such as the prechordal mesoderm, previously implicated in hypothalamic induction (*Dale et al., 1997*; *Dale et al., 1999*). To distinguish these possibilities, we isolated the emerging hypothalamus from other tissues at HH10 (*Figure 3G–I*). Acutely dissected explants were positive for pSMAD1/5/8/*SHH/FGF10* (*Figure 4J and K*), and negative for *FOXG1/ISL1/EMX2* (*Figure 3—figure supplement 2*), confirming the accuracy of dissection. *Figure 2—figure supplement 1AFigure 5Fu et al., 2017*. Similar explants cultured to a HH13 equivalent grew and generated domains that showed the same organisation as in vivo. *SHH* and *FGF10* now resolved into overlapping domains (*Figure 3L–L''*), and distinct but overlapping regions expressing *SIX6*, *SIX6/SHH* and *TBX2* were detected, as in vivo (compare *Figure 3M–M''* to *Figure 1N*). After culture to an HH24 equivalent, explants had grown extensively and showed discrete but overlapping domains of marker expression: an ISL1/SHH+ve domain, adjacent to a pSMAD1/5/8+ve domain, and then an EMX2+ve domain (*Figure 3N–N''''*). This pattern suggests that isolated explants, free from any possible influence of surrounding tissues, have grown to generate a spatial pattern that is similar to that in vivo, that is, composed of ISL1/SHH+ve anterior neurogenic cells, then pSMAD1/5/8+ve tuberal progenitor cells, then EMX2+ve mammillary cells (*Figure 3O–Q*). We conclude that, although BMP ligands from non-hypothalamic structures are required to initiate tuberal development (*Dale et al., 1997*; *Dale et al., 1999*), from HH10 onwards, hypothalamic neuroepithelium-intrinsic factors are sufficient to maintain tuberal hypothalamic regionalisation and neurogenesis.

## BMP signalling directs tuberal development during HH9-HH13

Our findings predict that exposure of HypFP cells to BMPs will lead to the generation of tuberal cells. To test this, we performed ex vivo studies, building on previous work that has delineated the spatial position of future hypothalamic regions and the time at which they are specified (*Kim et al., 2022*; *Ohyama et al., 2005*). We isolated explants from HH6 to HH8 HypFP regions that fate-map to either the tuberal or the mammillary hypothalamus, termed anterior and posterior explants, respectively (*Figure 4A and D*). After culture to a ~HH14 equivalent, anterior explants expressed tuberal progenitor markers (*SIX6/NKX2-1/SHH/TBX2*), and anterior-tuberal neuronal markers (*NR5A1, POMC*) (*Figure 4*, row B). By contrast, posterior explants did not express tuberal markers, but instead,

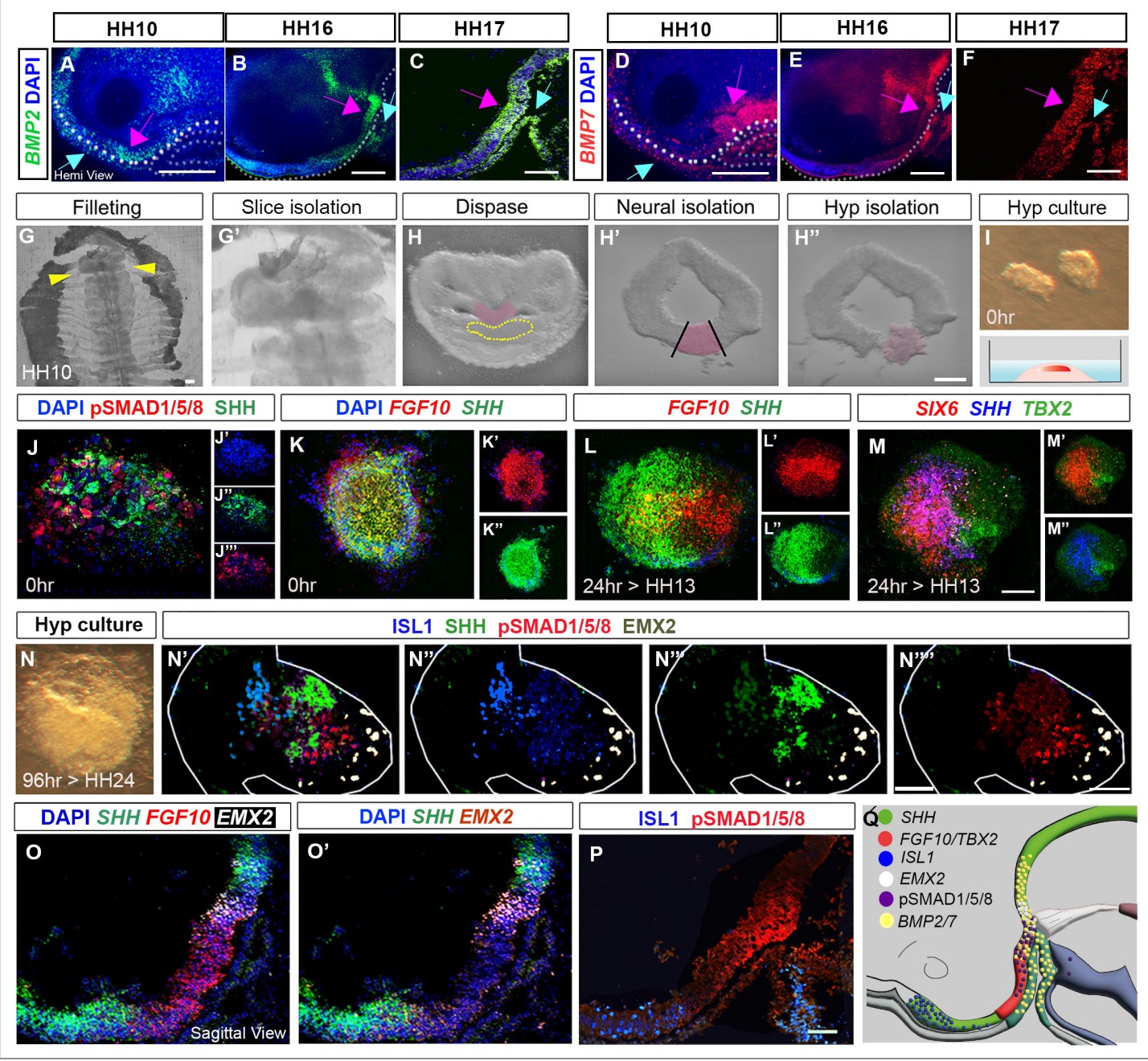

**Figure 3.** Organisation of tuberal domains through a neuroepithelial-intrinsic mechanism. (**A–F**) MIPs of hemi-dissected heads after HCR, showing expression of *BMP2* or *BMP7* at HH10, HH16 and HH17. Expression is detected in preplacodal ectoderm/Rathke's pouch (blue arrows) and the neuroectoderm (pink arrows). Dots in (A, B, D and E) outline basal neural tube. (**G–I**) Steps describing hypothalamic tissue isolation from an HH10 embryo: (**G, G'**) filleting, yellow arrows in (**G**) mark the position of the slice containing hypothalamic tissue, shown at higher power in (**G'**). (**H–H"**) Isolated slices containing hypothalamic tissue (pink) have a characteristic shape, and the prechordal mesoderm can be identified morphologically (dotted outline in **H**). After dispase treatment, neural tissue is isolated from surrounding tissues (**H'**), and hypothalamic tissue is excised (**H"**). (**I**) Isolated hypothalamic tissue is embedded in a 3-D collagen matrix. (**J–N""**) Sections through hypothalamic explants, at 0 hr, or cultured for 24 or 96 hr, analysed by HCR or immunolabelling. (**J**) At 0 hr, pSMAD1/5/8 and SHH are detected throughout the section (single channel views shown in (**J'–J"**), blue shows DAPI counterstain). (**K**) At 0 hr, *FGF10* and *SHH* are detected throughout the section single channel views shown in (**K', K"**). (**L–M**) After a 24-hr culture period, to the equivalent of HH13, *FGF10* and *SHH* begin to resolve ((L): single channel views shown in (**L', L"**)), and discrete domains of *SIX6*, *SHH* and *TBX2* are apparent ((**M**): double channel views shown in (**M', M"**)). (**N**) Brightfield image of explant cultured for 96 hr to the equivalent of HH24. (N'-N"") Same explant, after immunolabelling, shows organised expression of ISL1, SHH, pSMAD1/5/8 and EMX2 ((**N'**: double channel views shown in (**N"–N""**)). (**O–P**) Serial adjacent sagittal sections of HH19 embryos, analysed by HCR to show expression of *SHH*, *FGF10* and *EMX2* ((**O, O'**) shows same section without *FGF10*), or by immunolabelling to detect ISL1 and pSMAD1/5/8 (**P**). (**Q**) Schematic summarising the expression of *SHH*, *TBX2*, *FGF10*, *ISL1*,

*Figure 3 continued on next page*

*Figure 3 continued*

*EMX2*, pSMAD1/5/8 and *BMP2/7* at HH19. Scale bars: 100 µm. Each panel shows a representative image from a minimum of n=3 embryos or explants. HCR, hybridisation chain reaction; MIP, maximum intensity projection.

The online version of this article includes the following figure supplement(s) for figure 3:

**Figure supplement 1.** Tuberal progenitors arise in and around BMP-expressing neuroepithelial cells.

**Figure supplement 2.** Accurate isolation of hypothalamic tissue.

expressed *SHH/FOXA2* (*Figure 4*, rows C, E) markers that define the later floor plate and supramammillary hypothalamus (*Kim et al., 2022*). However, posterior explants that were exposed to 32 nM BMP2/7 upregulated tuberal markers, including *TBX2*, *SIX6*, *NR5A1* and *POMC* (*Figure 4F–H*). We conclude that BMP2/7 can induce HypFP cells to form tuberal cells.

To complement these studies, we tested whether a reduction in BMP signalling suppresses tuberal hypothalamic development in vivo. Implantation of beads, soaked with the BMP inhibitor Noggin (*Smith and Harland, 1992*) in the tuberal region of an HH9–HH10 embryo (*Figure 5A*) resulted in partial suppression of tuberal development at HH18. The $SIX6^{+ve}$ anterior domain was significantly reduced, there was no/little sign of the $TBX2^{+ve}$ posterior domain, and *SHH* was abnormally maintained in the region where *TBX2* is normally expressed (*Figure 5B–D*).

We reasoned that the presence of a few tuberal cells after exposure to Noggin could reflect either that Noggin does not persist in vivo, and/or that tuberal hypothalamic specification has already been initiated by HH10. Indeed, examination of Noggin-exposed embryos showed that pSMAD1/5/8 is downregulated only transiently (*Figure 5—figure supplement 1*). We therefore sought to eliminate BMP signalling at an earlier time point, and in a manner where the effects of Noggin could be sustained, turning again to the ex vivo explant assay. We first confirmed that anterior explants cultured

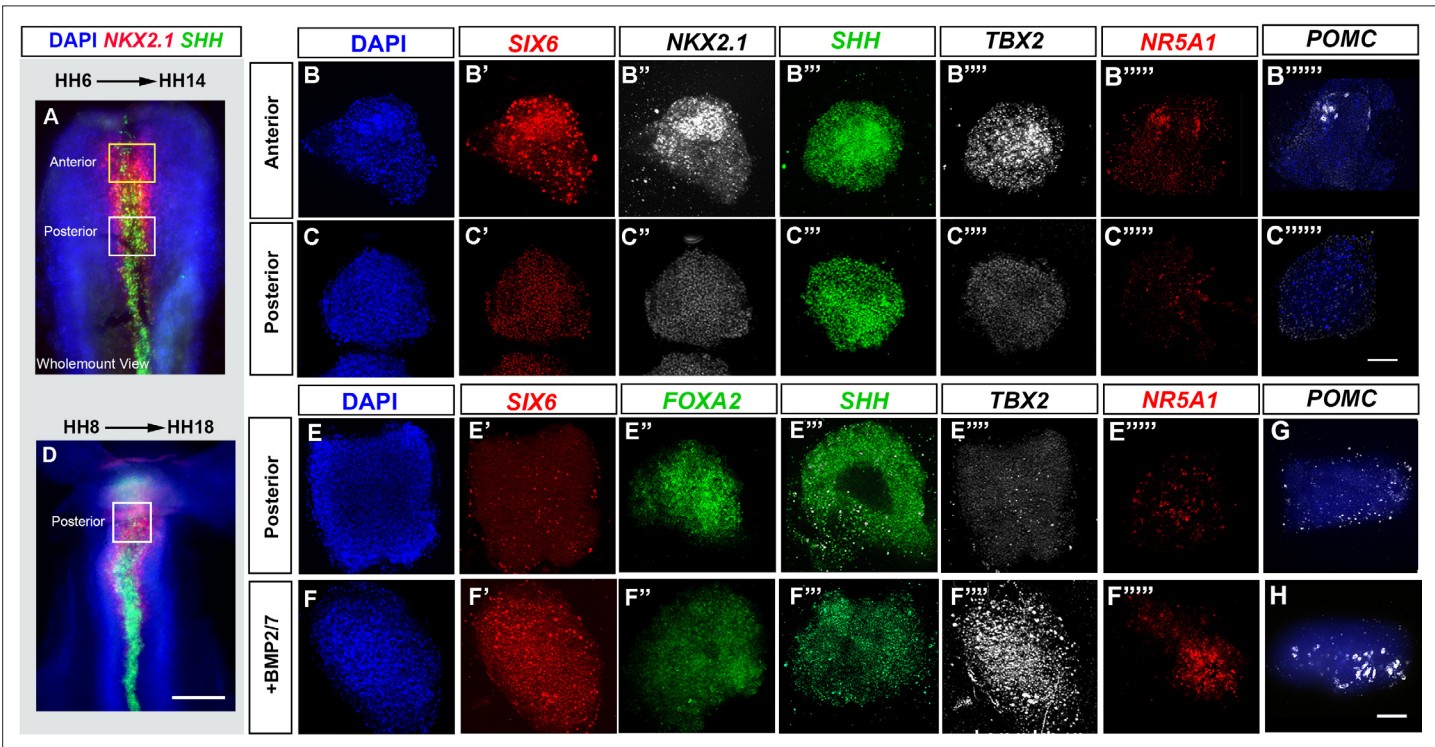

**Figure 4.** BMP promotes tuberal identity in HypFP cells. (**A, D**) HH6 (**A**) or HH8 (**D**) isolated neuroepithelia, after HCR to detect *NKX2-1* and *SHH*. Boxes show regions dissected for explant culture. (**B–B'''''', C–C'''''', E–E'''''', F–F'''''**) HH6 anterior or posterior explants cultured to a HH14 equivalent in control media (rows **B, C, E**) or 32 nM BMP2/7 protein (row **F**), and then processed through repeated rounds of multiplex HCR for *SIX6, SHH, TBX2/NKX2-1/NR5A1/POMC* or *SIX6/FOXA2/SHH/TBX2/NR5A1*. (**G–H**) HH8 posterior explants cultured to an HH18 equivalent in control media (G) or BMP2/7 (H) and processed by wholemount HCR for *POMC*. Scale bars: 100 µm. *NKX2-1, NR5A1, POMC* and *FOXA2* were analysed in a minimum of n = 3 explants/condition; *SIX6, SHH* and *TBX2* were analysed in a minimum of n = 11 HH6 explants/condition and n = 7 HH8 explants/condition. HCR, hybridisation chain reaction.

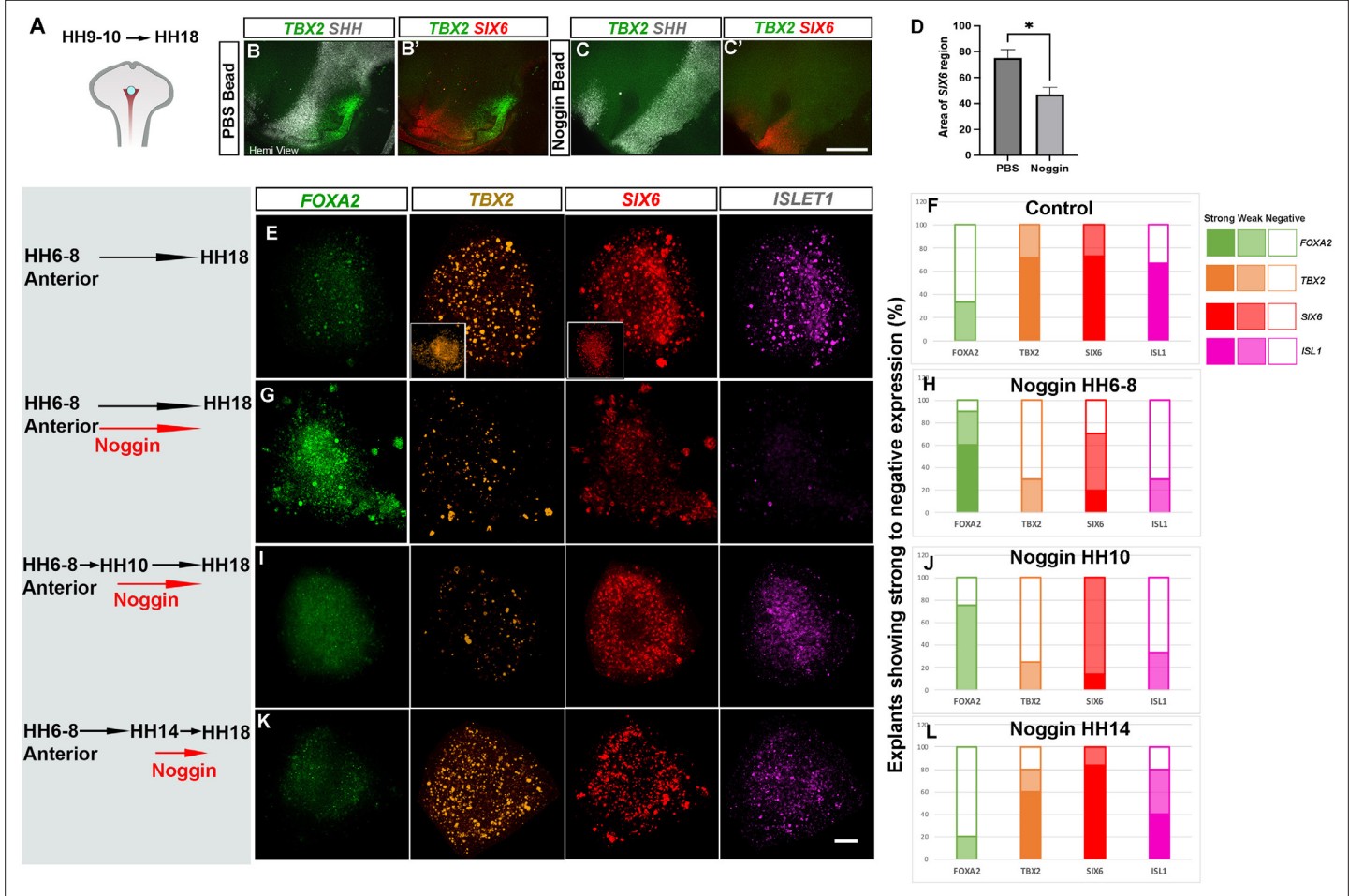

**Figure 5.** BMP signalling patterns anterior and posterior tuberal progenitors over time. (**A**) Schematic depicting position of bead implantation. (**B–C'**) Hemi-views of HH18 heads, analysed by HCR to detect *TBX2, SHH* and *SIX6* after PBS (**B, B'**) or Noggin (**C, C''**) bead implantation. (**D**) There is a significant decrease in the area of the *SIX6*$^{+ve}$ domain after exposure to Noggin p < 0.05*, unpaired t test; error bars show SEM ; n = 5 embryos/condition. (E, G, I, K) HH6–HH8 anterior explants, cultured to an HH18 equivalent, in the absence of Noggin (**E, F**), or after exposure to 300 ng/µl Noggin at the onset of culture (**G, H**), after culture to an HH10 equivalent (**I, J**) or after culture to an HH14 equivalent (**K, L**). Insets in row (**E**) show explants cultured to an HH10 equivalent. Each row shows a representative single explant, subject to HCR to detect *FOXA2/TBX2/SIX6/ISL1*. (**F, H, J, L**) Explants were scored for expression levels (strong expression >50% explant—e.g., *TBX2/SIX6/ISL* in row (**E**); weak expression <50% explant—e.g., *TBX2/SIX6* in row (**G**); no expression—e.g., *ISL1* in row (**G**)). n = 14 explants/condition (**F, H**); n = 4 explants/condition (**J, L**). Scale bars: 100 µm. HCR, hybridisation chain reaction.

The online version of this article includes the following source data and figure supplement(s) for figure 5:

**Source data 1.** Raw data for *Figure 5D*.

**Source data 2.** Raw data for *Figure 5F, H, J and L*.

**Figure supplement 1.** Transient reduction of pSMAD1/5/8 in vivo.

to an HH18 equivalent robustly express the tuberal markers *TBX2/SIX6*, and do not express the supra-mammillary/floor plate marker, *FOXA2* (*Figure 5E and F*). Further, we showed that many express the anterior neuronal precursor marker, *ISL1* (*Figure 5E and F*). In contrast, anterior explants cultured from the outset in 300 ng/µl Noggin showed little or no expression of either *SIX6, TBX2* or *ISL1*, but instead, robustly expressed *FOXA2* (*Figure 5G and H*). To determine whether ongoing BMP signalling is needed to maintain tuberal development, we inhibited BMPs at later time points within the same experimental setup. When anterior explants were cultured in control medium to the equivalent of HH10, both *SIX6* and *TBX2* were detected (*Figure 5E*, insets). However, if anterior explants were cultured in control medium to HH10, and then cultured with Noggin to HH18, *SIX6* and *ISL1* were now more robustly detected, but little/no *TBX2* was detected (*Figure 5I and J*). This suggests that

*TBX2*⁺ᵛᵉ cells generated by HH10 have progressed to anterior cells, and that anterior specification is well underway by HH10, but that a longer or later period of BMP exposure is required for the de novo generation of posterior progenitors. The addition of Noggin to anterior explants at 48 hr, which is the equivalent of ~HH14, had little effect, with *FOXA2*, *TBX2*, *SIX6* and *ISL1* detected similarly to the frequency and levels observed in controls (*Figure 5H–H''''*). This indicates that de novo specification of posterior progenitors has occurred by HH14.

## Excessive BMP levels disrupt tuberal progenitor tissue homeostasis

Our fate-mapping studies show that anterior tuberal neurogenic progenitors are derived from progenitors that experience BMP signalling only transiently. This raises the possibility that the cessation of BMP signalling is an important step in anterior tuberal specification, and we reasoned that prolonged/excessive exposure to BMPs may disrupt this. To directly test this, PBS-control or BMP2/7-soaked beads were implanted into the anterior-most region of the neural tube of HH10 embryos and embryos examined at HH13/14 using multiplex HCR. Ectopic BMPs reduced, or eliminated, *SIX6/SHH* ⁺ᵛᵉ anterior tuberal progenitors (white brackets in *Figure 6A–C''*). This supports the idea that the distancing of tuberal progenitors from the source of BMP in the posterior hypothalamus that occurs in vivo is a critical step in their specification to an anterior neurogenic state.

At the same time, ectopic BMPs altered the profile of the posterior tuberal domain (pink brackets in *Figure 6A''–C'*). Cells showed a 'mixed profile' (*SIX6/TBX2/SHH*⁺ᵛᵉ), similar to that detected at HH9. Additionally, *SHH* expression was marginally reduced in the anterior lateral hypothalamus, while *TBX2* expression was significantly increased (*Figure 6E*). No obvious changes in expression of the prethalamic marker *PAX6* were detected, but *FST*, which at this stage is restricted to a domain located just dorsal to the anterior intrahypothalamic diagonal (ID) (*Shimogori et al., 2010*), was eliminated (*Figure 6B and D*, white arrows). These observations extend previous studies that demonstrate that BMP signalling induces *TBX2* expression (*Manning et al., 2006*), but suggest additionally that excess BMP signalling prevents developing tuberal progenitors from developing beyond an early 'mixed' state.

To determine if the reduction in anterior neurogenic progenitors was maintained at later stages, a further set of embryos were examined at HH18, each taken through repeated rounds of multiplex HCR (probe combinations shown in *Figure 6—figure supplement 1A–D*). Exposure to BMP2/7 either reduced (n = 4) or eliminated (n = 2) anterior tuberal progenitors, as measured through a reduction or loss of *SIX6* and *SHH* (*Figure 6F–G'* and *K*; *Figure 6—figure supplement 1E–F*). Surprisingly, and in contrast to embryos examined at HH13, BMP-treated embryos also showed a reduction or loss of posterior tuberal progenitors, reflected in the decreased or absent expression of *TBX2* and *FGF10* (*Figure 6H–I'*; *Figure 6—figure supplement 1G, G'*). By contrast, *FOXA2* expression in the floor plate and supramammillary hypothalamus appeared normal (*Figure 6J and J'*; *Figure 6—figure supplement 1H, H'*). Additional rounds of multiplex HCR confirmed the reduction or loss of the anterior and posterior tuberal progenitor territories as well as the ID, as judged by reduction or loss of *RAX*, *NKX2-1*, *NKX2-2* and *FST* expression (*Figure 6L–O'* and *Q*; *Figure 6—figure supplement 1I–J'*). These analyses confirmed that posterior hypothalamic regions were barely affected: expression of the supramammillary and mammillary markers *PITX2* and *EMX2* were maintained (*Figure 6—figure supplement 1K–L'*). The telencephalic marker *FOXG1*, likewise, was unaffected, maintaining its relation to the optic stalk opening (*Figure 6P and P'*). The reduction/loss of tuberal progenitor domains was accompanied by marked changes in anterior ventral forebrain morphology: the optic stalk failed to narrow, resulting in a gaping entrance to the optic stalk and copious *SIX6/RAX/PAX6*⁺ⁱᵛᵉ eye tissue within (*Figure 6F', F', L, L', O, O', R, R' and W*). We conclude that excessive BMP exposure leads ultimately to a loss of expression of molecular markers of both anterior and—unexpectedly—posterior tuberal progenitors.

We next examined expression of the postmitotic neuronal precursor marker *ISL1* to ask how the reduction in anterior and tuberal progenitors impacts neurogenesis. *ISL1* expression was absent (*Figure 6—figure supplement 1M, M'*), or weaker/reduced (*Figure 6S, S'*) after exposure to BMPs, in a manner that correlated with the loss or reduction of *SIX6/SHH*⁺ᵛᵉ anterior progenitors. Further analysis showed that in embryos with a partial loss of the tuberal hypothalamus, expression of the Notch target gene *HES5* was dramatically reduced (*Figure 6T and T'*). Notch maintains the progenitor state in the chick hypothalamus (*Hamdi-Rozé et al., 2020*; *Place et al., 2022*; *Ratié et al., 2014*, *Ratié*

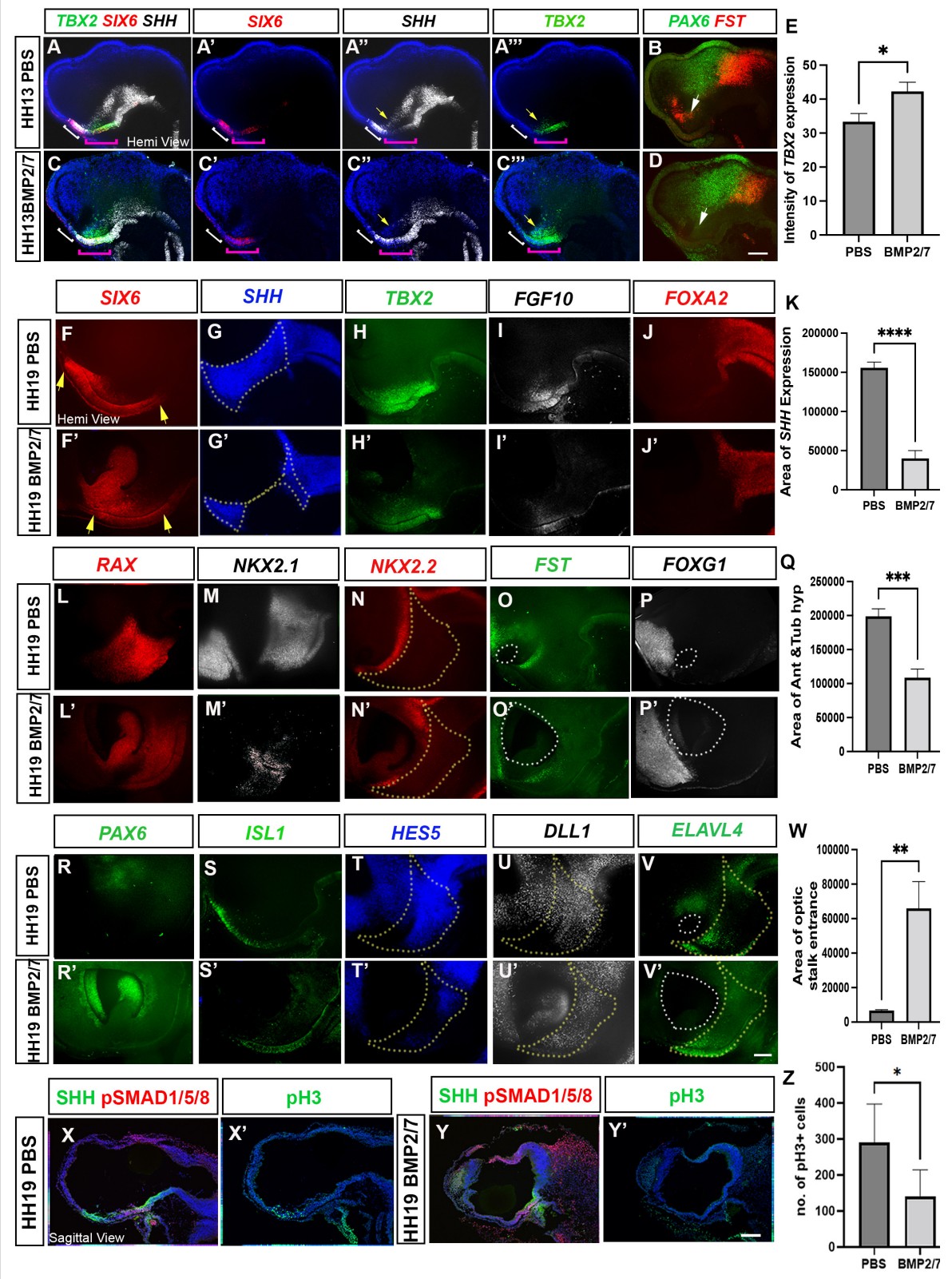

**Figure 6.** Ectopic BMP exposure reduces anterior, then posterior tuberal progenitors. (**A–D**) MIPs of hemi-dissected head of HH13/14 embryo taken through two rounds of multiplex HCR analysis, after implantation of PBS bead (**A**) (single channel views in **A'–A'''**, **B**) or BMP2/7-soaked bead (**C**) (single channel views in **C'–C''', D**). In (**A–C'''**), white bracket shows the anterior tuberal domain, absent after BMP2/7 exposure; pink bracket shows the posterior tuberal domain; yellow arrows show anterior ID. In (**B, D**), white arrows point to *FST⁺ᵛᵉ* cells, absent after exposure to BMP2/7. (**E**) Graph

*Figure 6 continued on next page*

*Figure 6 continued*

showing a significant increase in intensity of *TBX2* following BMP2/7 bead implantation p < 0.05, unpaired t test; error bars show SEM; n = 3 embryos/condition. (**F–J'; L–P'; R–V'**) Hemi-dissected heads of HH19 embryos after implantation of PBS bead (**F–V**) or BMP2/7-soaked bead (**F'–V'**). Each panel shows MIP of single channel views (multiplex views shown in *Figure 6—figure supplement 1*). In (**F, F'**), yellow arrows indicate the length of the *SIX6$^{+ve}$* domain. Dotted outlines in (**G, G'**) show area of *SHH* expression, measured in (K). Yellow dotted outlines in(**N, N', T, T', U, U', V, V'**) show area of anterior and tuberal hypothalamus, measured in (Q). White dotted outlines in (O, O' P, P' and V, V' ) show area of optic stalk entrance, measured in (**W**). (**K, Q**) There is a significant decrease in the area of *SHH* expression, p < 0.0001****, unpaired t test and a significant decrease in the area of anterior and posterior tuberal hypothalamus, p < 0.0004***, unpaired t test between PBS and BMP2/7 bead-implanted embryos. n = 6 embryos/condition; error bars show SEM. (**W**) There is a significant increase in the area of optic stalk entrance, p < 0.001**, unpaired t test. n = 4 embryos/condition; error bars show SEM.. (**X–Y'**) Serial adjacent sagittal sections of HH17 embryos, immunolabelled to detect expression of SHH, pSMAD1/5/8 or the M-phase marker phosphoH3 (pH3) after implanting PBS bead (**X, X'**) or BMP2/7-soaked bead (**Y, Y'**) at HH10. (**Z**) Graph showing significant decrease in the number of pH3 positive M-phase progenitors (p < 0.05*, unpaired t test; n = 5 embryos/condition; error bars show SD). Scale bars: 100 µm. HCR, hybridisation chain reaction; MIP, maximum intensity projection.

The online version of this article includes the following source data and figure supplement(s) for figure 6:

**Source data 1.** Raw data for *Figure 6E*.

**Source data 2.** Raw data for *Figure 6K, Q and W*.

**Source data 3.** Raw data for *Figure 6Z*.

**Figure supplement 1.** Ectopic BMPs reduce or eliminate tuberal cells.

*et al., 2013*; *Ware et al., 2016*), so we asked whether the loss of *HES5* expression was associated with ectopic neurogenesis, examining expression of the neurogenic progenitor marker *DLL1*, and the neural precursor marker *ELAVL4*. Neither, however, was obviously different in the residual hypothalamic domain of BMP-treated embryos in comparison to controls (*Figure 6U–V'*). A possible interpretation is that an increase in neurogenesis may be balanced out by the presence of fewer progenitor cells (*Figure 6F–K and Q*). We further explored this idea by examining proliferation in control versus BMP-treated embryos. Supporting this possibility, BMP-treated embryos showed a significant reduction in M-phase progenitors in the tuberal hypothalamus (*Figure 6X–Z*). We conclude that excessive BMP levels acutely inhibit anterior tuberal specification, and over time result in more widespread disruption, including the loss of posterior progenitors.

## Tuberal progenitors change over time

Our findings suggest that distinct spatial domains of the tuberal hypothalamus are generated sequentially as the result of transient BMP signalling. This implies that pSMAD1/5/8$^{+ve}$ progenitors may show dynamic patterns of gene expression that control their ability to generate distinct subsets of tuberal cells. Throughout the period HH10-HH17, pSMAD1/5/8$^{+ve}$ nuclei lie within/overlap with cells that express *FGF10* (*Figure 7A*; *Figure 3O–Q*), allowing us to use *FGF10* expression as a proxy for cells expressing pSMAD1/5/8. To identify genes that are dynamically expressed, we digitally isolated *FGF10$^{+ve}$* cells from our HH8-HH20 scRNA-Seq data set (*Figure 7B*; *Kim et al., 2022*). This analysis showed differential expression of a number of known developmental regulators between early (more anterior) and late (more posterior) *FGF10$^{+ve}$* cells (*Figure 7C*). *CHRD (Chordin)*, *BMP2/7* and *SST* showed enriched expression in early stage (HH8–HH10) tuberal progenitors, whereas the retinoic acid binding protein *CRABP1*, *WNT5A,* and the growth factor *CTGF* showed enriched expression in later-stage tuberal progenitors. The WNT/FGF pathway inhibitor *SHISA2* was expressed in early, but not late, tuberal progenitors, and expression of the canonical WNT mediator (and adherens junction component) *CTNNB1* (*β-catenin*) increased over time. Changes in expression levels of the NOTCH signalling pathway modulator *LFNG*, and its target genes *HES5* and *HES5-like* indicated a gradual increase in NOTCH signalling over time. Likewise, the transcription factors *ONECUT1*, *ID1*, *ID2, ID4, SOX8* and *SOX9* showed enriched expression in later-stage *FGF10$^{+ve}$* tuberal progenitors. HCR analyses confirm the upregulation of *ID2* and *ID4* in later-stage (posterior) tuberal progenitors (*Figure 7D–I*; *Place et al., 2022*). Similar transcriptional changes are detected more widely, perhaps indicating that these occur in response to a general tissue maturation. We conclude that both abrupt (e.g., *BMP2/7, SHISA2*) and gradual changes in gene expression are detected, conferring a distinct transcriptional profile upon *FGF10$^{+ve}$* cells at each stage sequenced.

Our ex vivo experiments suggested that anterior and posterior tuberal progenitors are specified by HH14 (*Figure 5*). Additional ex vivo experiments, isolating posterior tuberal progenitors at HH15 and

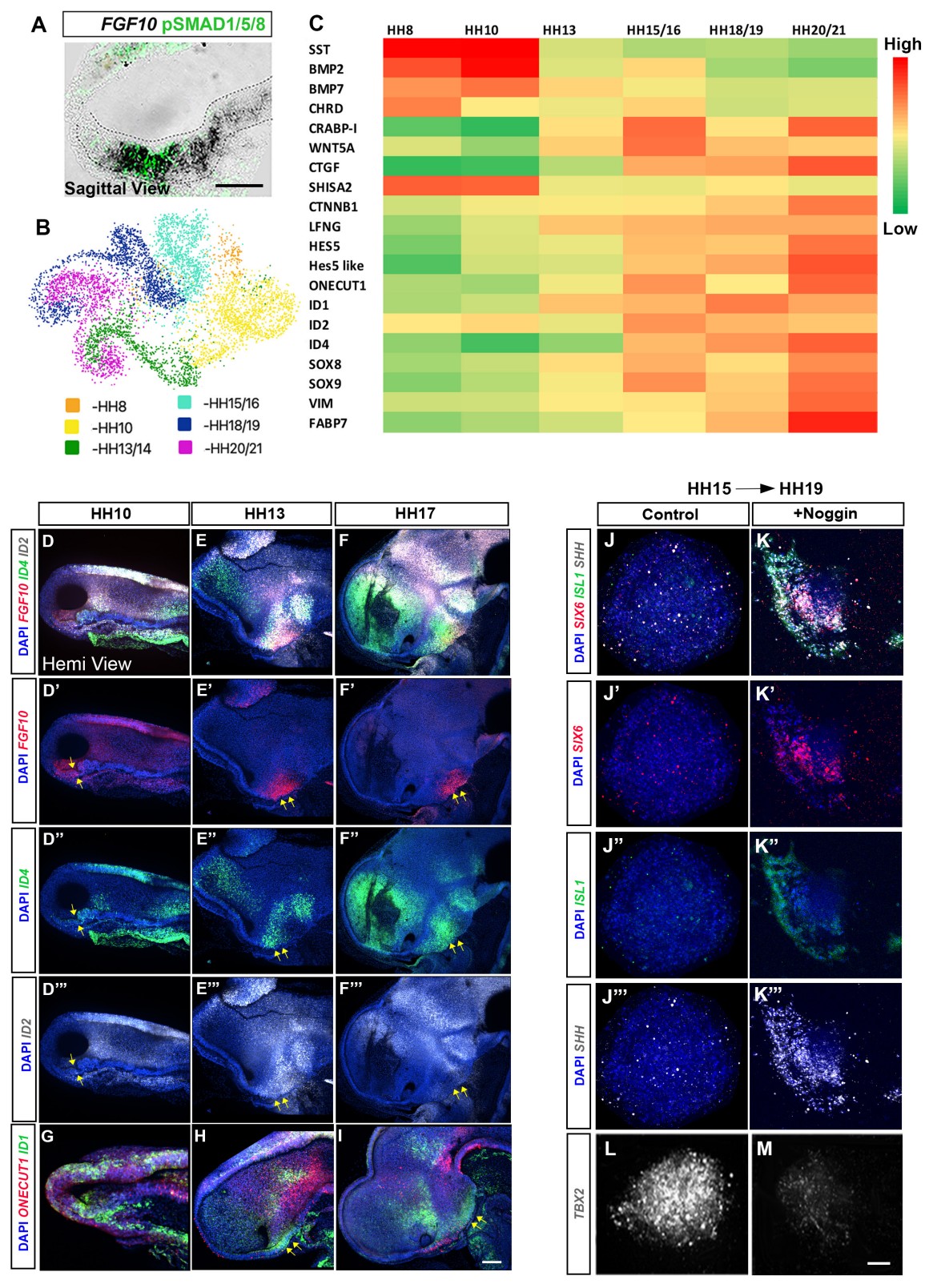

**Figure 7.** Dynamic transcriptional changes in *FGF10*-expressing tuberal progenitors. (**A**) Sagittal sectionof HH10 embryo, analysed by chromogenic in situ hybridisation and then by immunohistochemistry to detect *FGF10* and pSMAD1/5/8. (**B**) UMAP plot showing the distribution of *FGF10*⁺ᵛᵉ tuberal progenitors profiled in a previous scRNA-Seq study (*Kim et al., 2022*). (**C**) Heat map showing upregulated and downregulated genes between stages HH8 and HH20/21. (**D–I**) MIPs of hemi-dissected heads of HH10 (**D, G**), HH13 (**E, H**) and HH17 (**F, I**) embryos after multiplex HCR for *FGF10/ID4/ID2*

*Figure 7 continued on next page*

*Figure 7 continued*

(**D–F"**) or *ONECUT1/ID1* (**G–I**). Yellow arrows point to *FGF10+ve* domain. (**J–M**) MIPs of explants dissected from the *FGF10+ve* domain at HH15 i.e., region shown by arrows in (**E', F'**) and cultured to an HH19 equivalent alone (**J–J''', L**) or with Noggin (**K–K''', M**) analysed by HCR for *SIX6/ISL1/SHH/TBX2*. n = 3 explants/condition. Scale bars: 100 μm. HCR, hybridisation chain reaction; MIP, maximum intensity projection.

culturing them to HH17 show that Noggin eliminates TBX2 and promotes continued expression of anterior markers (*Figure 7J–M*). Therefore, BMPs actively *repress* the anterior tuberal program after HH15, and this switch in competence coincides with the transcriptional onset of a large set of transcription factors (*Figure 7C*). Upregulation of *SOX8/9* and *ID* family gene expression has been previously reported in late-stage neuronal progenitors in mammals (*Clark et al., 2019*; *Lyu et al., 2021*; *Sagner et al., 2021*; *Telley et al., 2019*), raising the possibility that the expression of these genes in later posterior tuberal progenitors is associated with neurogenesis. Alternatively, this upregulation may be directly linked to the transition of posterior tuberal progenitors from a neuroepithelial to radial glial-like state, a process that is regulated by SOX9 in the developing spinal cord (*Scott et al., 2010*). In support of this, the radial glial markers *VIM* and *FABP7* are upregulated in *FGF10+ve* progenitors at HH20/21 (*Figure 7C*). These molecular changes, therefore, may reflect the onset of the transition of *FGF10+ve* posterior tuberal neuroepithelial progenitors to the *FGF10+ve* radial glial-like tanycytes that characterise the late-embryonic and adult tuberal hypothalamus (*Goodman et al., 2020*; *Haan et al., 2013*; *Robins et al., 2013*; *Yoo et al., 2021*).

## Discussion

Building on earlier work, we set out to explore the mechanisms underlying formation of the tuberal hypothalamus—a region which harbours neurons and glia that govern hunger, energy balance, reproduction, response to stress, and a range of social, sexual, affiliative and emotional behaviors (*Saper and Lowell, 2014*; *Swaab, 2003*). We show that tuberal hypothalamic progenitors are laid down in a topological sequence, from anterior to posterior. In many regions of the CNS, including the cortex and retina, waves of neuronal precursors are generated and laid down in a topological sequence as stem-like/progenitor cells undergo temporal switches in developmental potential (*Heavner and Pevny, 2012*; *Kawaguchi, 2019*). Our studies show, by contrast, that in the tuberal hypothalamus, spatial pattern emerges through a series of transient progenitors, as a wave of BMP signalling sweeps from anterior to posterior through HypFP cells. In addition, we uncover candidate molecular regulators of the transition of tuberal progenitors to neurogenesis and gliogenesis.

### Specification of the tuberal neurogenic hypothalamus from hypothalamic floor plate-like cells

Several lines of evidence support the conclusion that tuberal progenitors are generated from a series of HypFP cells. First, tuberal progenitor markers are first detected co-incident with pSMAD1/5/8 in aHypFP cells. As tuberal progenitors are generated, and the tuberal hypothalamus lengthens, pSMAD1/5/8 consistently straddles the border of aHypFP and posterior-most tuberal progenitor cells. Second, gain-of-function studies demonstrate that ectopic BMP signalling can induce tuberal hypothalamic progenitors from HypFP cells that normally fate-map to the posterior (mammillary) hypothalamus. Conversely, the ability of aHypFP cells to give rise to tuberal progenitors can be almost completely prevented by exposure to the BMP inhibitor, Noggin. Finally, fate-mapping studies at HH10 demonstrate the spatial relationship between HypFP cells and tuberal progenitors: anterior-most tuberal progenitors are generated from aHypFP cells, with more posterior tuberal progenitors generated from more posterior HypFP cells.

Our fate-mapping studies support the idea that a steady stream of tuberal progenitors is laid down in the wake of a travelling active BMP zone, that passes from anterior to posterior through HypFP cells. We observe that neighbouring cells retain their relative positions along the anterior-posterior midline such that tuberal progenitors are laid down in an anterior-to-posterior order. Our studies support our previous scRNA Seq studies and suggest a sequence of differentiation from HypFP, to posterior tuberal progenitor, anterior tuberal progenitor, anterior tuberal neurogenic progenitor, anterior neural precursor and finally to mature tuberal neuron. While we cannot exclude that there may be tuberal progenitors that have a different history, and do not require BMP signalling (see below), our results

indicate that most, if not all, tuberal *SIX6*[+ve] anterior cells transiently activate BMP Smads and express *TBX2*, and are derived from *FOXA2*-expressing HypFP cells. Building on our previous studies (*Fu et al., 2017*; *Kim et al., 2022*), we show, moreover, that as tuberal progenitors are generated, the anterior region expands dramatically, resulting in the rapid separation of the telencephalon and the floor plate.

Our studies address a general question in development—as to how a limited set of extrinsic signals can robustly instruct diverse neural progenitors. The wave of BMP signalling could provide a mechanism for temporal patterning of tuberal neurogenic progenitors, with the time at which HypFP cells are exposed to BMPs determining their future identity.

We detect changes in the gene expression profile of *FGF10*-expressing tuberal hypothalamic progenitors between HH10 and HH13, providing candidates for the molecular players behind the changing competence of these cells over time (*Figure 7*). The upregulation of *SOXE, ID* and *ONECUT* family transcription factors, as well as differential expression of the *SHH, WNT, FGF* and *NOTCH* pathway components, could potentially determine the specification of distinct tuberal neuronal and glial subtypes as they encounter BMP signalling.

An intriguing question is how the wave of BMP signalling is transmitted backwards through HypFP cells. Many studies have shown the dependence of hypothalamic induction on the ventrally located prechordal mesoderm, which expresses BMP ligands (*Mathieu et al., 2002*; *Ellis et al., 2015*; *Dale et al., 1997*; *Dale et al., 1999*). However, our ex vivo experiments indicate that, from as early as HH6, the tuberal differentiation programme can be sustained in the absence of any extrinsic sources of BMPs, suggesting the importance of a neuroepithelial-intrinsic mechanism after this time.

## A BMP-SHH balance governs tissue homeostasis of tuberal progenitors

Many previous studies point to the importance of SHH in tuberal neurogenesis (*Carreno et al., 2017*; *Corman et al., 2018*; *Fu et al., 2017*; *Shimogori et al., 2010*; *Szabó et al., 2009*; *Wang et al., 2015*). This, and the fact that SHH and BMP signals/signalling often act antagonistically in neural tube patterning (*Ulloa and Briscoe, 2007*) are counter-intuitive to our finding that BMP signalling is critical to tuberal development. How might these findings be reconciled?

First, one possibility is that BMP plays a role in the initial specification of tuberal progenitors by inhibiting FST. Here, we report that BMPs can suppress *FST* expression, and previously we have shown that FST inhibits hypothalamic induction (*Kim et al., 2022*). Second, our experiments make it clear that BMP signalling is needed only for the initial specification of *TBX2*[+ve] posterior tuberal progenitors (*Manning et al., 2006*) and must be switched off for these to progress to the anterior tuberal programme: BMP inhibition in HH15 explants led to the loss of posterior, and the gain of anterior tuberal markers; conversely, prolonged exposure of nascent tuberal progenitors to BMPs at HH10 compromised anterior tuberal development. We suggest that by ensuring the sustained generation of appropriate numbers of *SIX6/SHH* progenitors, BMP signalling supports the development of a continued source of SHH-producing cells that sustain the tuberal neurogenic programme. Without the continued specification of posterior tuberal progenitors, a smaller *SIX6/SHH*[+ve] anterior tuberal domain may be inadequate to support the tuberal programme (*Figure 8*).

Intriguingly, while posterior tuberal progenitors give rise to anterior tuberal progenitors, our results support previous observations that anterior progenitors generate signals that maintain posterior progenitors: the loss or reduction of *SHH*[+ve] anterior progenitors (in response to excessive BMP signalling) correlates, at HH18, with a loss or reduction of posterior tuberal progenitors. Chick studies have previously shown that a transient reduction in SHH signalling disrupts *FGF10*[+ve] posterior progenitors (*Fu et al., 2017*). We therefore propose that, as in the developing cerebellum, SHH from more differentiated cells promotes proliferation of earlier-stage progenitors (*Lewis et al., 2004*; *Wechsler-Reya and Scott, 1999*; *Figure 8*). Supporting this, the loss/reduction of tuberal markers seen following prolonged BMP exposure is accompanied by a significant decrease in proliferating cells.

Finally, despite the loss of tuberal progenitors in BMP-exposed embryos, neurogenic (*DLL1/ASCL1*) and tuberal neuronal markers (*ISL1/NR5A1/POMC*) were detected. This was furthermore associated with a reduction in *HES5*, suggesting that Notch signalling is also compromised in these embryos. This would be expected to result in premature neuronal differentiation of hypothalamic neurons (*Aujla et al., 2013*; *Place et al., 2022*; *Ratié et al., 2013*), providing a possible explanation for the presence of these markers even as progenitor development is heavily compromised. Our results overall

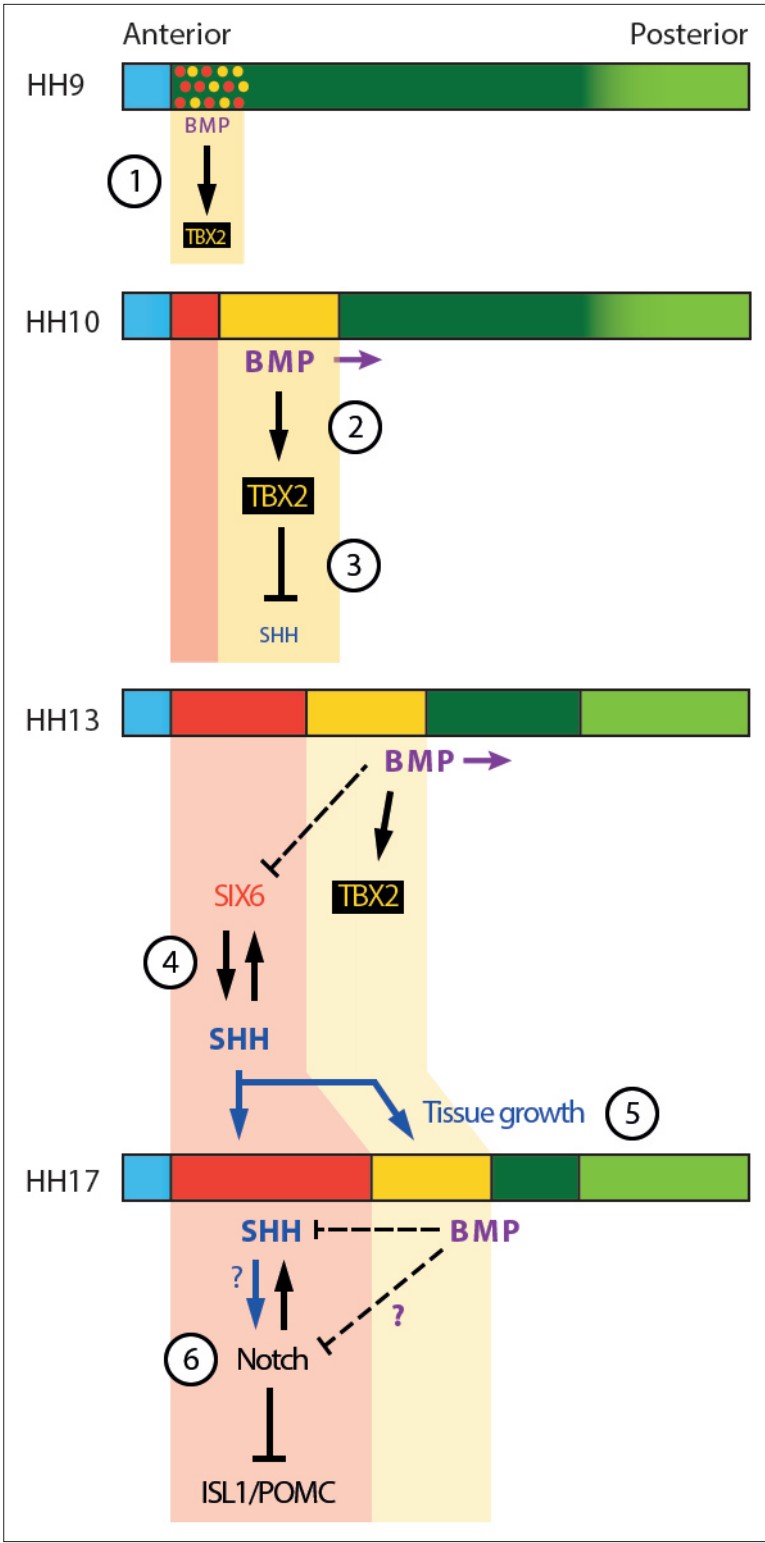

**Figure 8.** Schematic depicting selected signalling/transcription factor interactions in developing tuberal neuroectoderm. (1) The onset of BMP signalling in HH8/9 anterior neuroectoderm starts to induce *TBX2* in HypFP cells. (2) BMP signalling starts to move posteriorly within the neuroepithelium (purple arrows), promoting *TBX2* expression in its vicinity. (3) TBX2 in turn represses *SHH*, which starts to be downregulated. (4) A mutually reinforcing feedback loop between SIX6 and SHH may help to establish/maintain robust expression of both factors in the anterior tuberal domain. Note that BMP signalling represses *SIX6* expression after HH15, whether

*Figure 8 continued on next page*

*Figure 8 continued*

directly or indirectly (dotted line). (5) SHH produced by anterior tuberal progenitors may feed back onto posterior tuberal progenitors to sustain their growth, and that of the anterior domain. (6) By unknown mechanisms (question marks), fine-tuning of Notch signalling ensures the correct balance of proliferation versus neurogenesis in the tuberal hypothalamus. Horizontal coloured bars represent HH8–HH17 neuroectoderm; colouring is the same as *Figure 1O-S*, *Figure 2P*. See *Figure 1O-S* for corresponding gene expression profiles.

suggest that the delicate balance of FST, BMP, SHH, FGF and Notch signalling ensures the correct balance between progenitor specification, proliferation and neurogenesis, laying the groundwork for the further development of this highly complex hypothalamic region (*Figure 8*).

## Establishment of a stem cell-like zone in the tuberal hypothalamus?

Our ex vivo studies show that tuberal patterning is largely complete by HH14. Our in silico bioinformatic analysis indicates that by HH20/21, posterior tuberal progenitor cells undergo a significant change, upregulating *VIM* and *FABP7*, markers of radial glial cells. Elsewhere in the CNS, the transition of neuroepithelial cells to radial glial cells coincides with the onset of stem-like character, and potentially, this is true also in the posterior tuberal hypothalamus. In all vertebrates examined to date, specialised radial glial-like cells, termed tanycytes, form the critical constituents of a stem cell-like niche that is present in the adult hypothalamus, situated in the median eminence, adjacent to the pituitary gland. Studies in the mouse have shown that adult *FGF10*-expressing tanycytes retain stem and progenitor-like activities (*Goodman et al., 2020*; *Haan et al., 2013*; *Robins et al., 2013*). We propose that the molecular changes that we detect in posterior tuberal progenitor cells over HH13–HH21 govern the transition from neurogenic *FGF10*-expressing neuroepithelial cells to *FGF10*-expressing radial glia, and the onset of stem-like characteristics in these cells. This work therefore indicates candidate regulators that govern development of the adult hypothalamic stem cell niche.

## Materials and methods

### Chick collection

Fertilized Bovan Brown eggs (Henry Stewart & Co., Norfolk, UK) were used for all experiments. All experiments were performed according to relevant regulatory standards (University of Sheffield). Eggs were incubated and staged according to the HH chick staging system (*Hamburger and Hamilton, 1992*).

### Chicken HCR

HH stage 8–20 embryos were harvested and fixed in 4% paraformaldehyde. For wholemounts, the embryos were dehydrated in a methanol series and stored at –20°C. HCR v3.0 was performed on embryos and cryosections using reagents and protocol from Molecular Instruments, Inc Samples were preincubated with a hybridisation buffer for 30 min and the probe pairs were added and incubated at 37°C overnight. The next day samples were washed four times in the probe wash buffer, two times in the 5× SSC buffer and preincubated in amplification buffer for 5 min. Even and odd hairpins for each gene were snap-cooled by heating at 95°C for 90 s and cooling to RT for 30 min. The hairpins were added to the samples in amplification buffer and incubated overnight at RT in the dark. Samples were then washed in 5× SSC and counterstained with DAPI. For multiplexing, after imaging with the first set of probes, wholemount samples were treated with DNAase (0.2 U/µl, Roche) overnight, washed three times in 30% formamide and 2× SSC and three times in 2× SSC. Slides were then preincubated with the hybridisation buffer, the next set of probes was added, and the process was repeated.

### Immunohistochemistry

Embryos and explants were analysed by immunohistochemistry according to standard techniques (*Manning et al., 2006*). Embryos or cryosectioned sections were analysed using the following antibodies anti-pSMAD1/5/8 (1:500, Cell Signaling Technology, 9511), anti-FOXA2 (1:50, DSHB, 2C7), anti-LHX3 (1:50, DSHB, 64.4E12/67.4E12), anti-SHH (1:50, DSHB, 5E1), anti-ISLET1 (1:50, DSHB, 4D5.65) and anti-PH3 (1:1000, Cell Signaling Technology, 06-570). Secondary antibodies (1:500,

Jackson ImmunoResearch) were conjugated with anti-Alexa 488 or 594. Images were taken using a Zeiss Apotome.

## Single-cell analysis

We analysed our previous scRNA-Seq data for the developing chicken hypothalamus (*Kim et al., 2022*). Hypothalamic cells expressing *FGF10* were subsetted and used in this study. Differential genes expressed in *FGF10*[+ve] cells across developmental time points were plotted on the heatmap.

## Neural tube isolation, hypothalamic tissue dissection and explant culture

Neural tubes were isolated from surrounding tissue by dispase treatment, as previously described (*Ohyama et al., 2005*). Explants of prospective hypothalamus were isolated from HH6 to HH8 embryos after dispase treatment as previously described (*Ohyama et al., 2005*). For isolation of hypothalamus at HH10, heads were first filleted, and slices containing hypothalamic tissue isolated on the basis of their characteristic shape, and the morphologically distinct underlying prechordal mesoderm, were then subject to dispase treatment. Explants of *FGF10*[+ve] tuberal region were isolated from HH15 embryos after dispase treatment. Where relevant, explants were treated with recombinant BMP2/7 heterodimers (32 nM, R&D Systems, Cat no. 3229-BM-010) or Noggin (300 ng/µl, R&D Systems, Cat no. 1967-NG) for 24 or 48 or 72 or 96 hr. Explants were processed for in situ HCR or immunohistochemistry.

## In vivo manipulation of BMP signalling

Affi-Gel beads (Bio-Rad, 153-7301) were soaked in BMP2/7 (32 nM) or Noggin (300 ng/µl) for 2 hr at 4°C and grafted into the anterior region of HH10 embryo. Embryos were allowed to develop to HH13 or HH17 (42 hr), and processed for HCR and immunohistochemistry.

## Fate-Mapping

HH10 embryos were windowed and a dorsal incision was made to access the ventral midline/floor-plate. To aid visualisation, Coomassie Blue was diluted in L15 at 0.5 µl/ml and injected under the embryo using a 23G needle and syringe. About 50 µg tubes of CellTracker CM-DiI Dye (Invitrogen, Cat no. C7000) were diluted in 30 µl ethanol, and 2 µl loaded into a fine glass needle, previously pulled to a sharp point using a needle puller. DiI was injected into embryos by hand using a Parker Picospritzer II set to 15 psi for 10–20 ms. Embryos were allowed to develop to HH17 and then harvested, fixed and processed for HCR and immunohistochemistry. DiI/DiO fate-mapped HH10 Dispase-isolated neural tubes were mounted on collagen beds on glass-bottomed dishes (Thermo Fisher Scientific, 50-305-806) and cultured for 24 hr at 37 and imaged every 6 hr.

## Image acquisition and quantification

Fluorescent images were taken on a Zeiss Apotome 2 microscope with Axiovision software (Zeiss) or Leica MZ16F microscope or Olympus BX60 with Spot RT software v3.2 or Nikon W1 Spinning Disk Confocal with Nikon software. Images were acquired using a 4× (Leica), 10× (Leica and Zeiss) and 10× and 20× objective Zeiss and Nikon microscope. Images were processed and digitally aligned using ImageJ (FIJI) and Adobe Photoshop 2021. Unpaired t tests and one-way analysis of variance were run on GraphPad Prism 9, and $p < 0.05$ was taken as significant. Mean ± SEM are plotted.

## Acknowledgements

The authors thank Transcriptomics and Deep Sequencing Core (Johns Hopkins) for sequencing scRNA-Seq libraries. This work was supported by the Wellcome Trust (212247/Z/18/Z) to MP.

# Additional information

## Funding

| Funder | Grant reference number | Author |
|---|---|---|
| Wellcome Trust | 212247/Z/18/Z | Marysia Placzek |

The funders had no role in study design, data collection and interpretation, or the decision to submit the work for publication. For the purpose of Open Access, the authors have applied a CC BY public copyright license to any Author Accepted Manuscript version arising from this submission.

## Author contributions

Kavitha Chinnaiya, Data curation, Software, Formal analysis, Validation, Investigation, Visualization, Methodology, Project administration, Writing – review and editing; Sarah Burbridge, Aragorn Jones, Data curation, Formal analysis, Validation, Investigation, Visualization, Methodology, Writing – review and editing; Dong Won Kim, Elsie Place, Data curation, Software, Formal analysis, Validation, Investigation, Methodology, Writing – review and editing; Elizabeth Manning, Data curation, Software, Formal analysis, Validation, Investigation, Visualization, Methodology, Writing – review and editing; Ian Groves, Formal analysis, Validation, Investigation, Methodology, Writing – review and editing; Changyu Sun, Data curation, Software, Formal analysis, Validation, Investigation; Matthew Towers, Conceptualization, Writing – review and editing; Seth Blackshaw, Conceptualization, Resources, Software, Formal analysis, Supervision, Funding acquisition, Validation, Project administration, Writing – review and editing; Marysia Placzek, Conceptualization, Resources, Data curation, Software, Formal analysis, Supervision, Funding acquisition, Validation, Investigation, Visualization, Methodology, Writing – original draft, Project administration, Writing – review and editing

## Author ORCIDs

Kavitha Chinnaiya 
Aragorn Jones 
Dong Won Kim 
Matthew Towers 
Seth Blackshaw 
Marysia Placzek 

## Decision letter and Author response

Decision letter https://doi.org/10.7554/eLife.83133.sa1
Author response https://doi.org/10.7554/eLife.83133.sa2

# Additional files

## Supplementary files

- MDAR checklist

- Source data 1. Source data provides accession numbers for the HCR probes used in this study.

## Data availability

All data generated or analysed during this study are included in the manuscript and supporting file. All chick single-cell RNA-Seq data have been deposited at GEO (GSE171649) and are publicly available.

The following previously published dataset was used:

| Author(s) | Year | Dataset title | Dataset URL | Database and Identifier |
|---|---|---|---|---|
| Kim DW, Place E, Chinnaiya K, Manning E, Sun C, Dai W, Ohyama K, Burbridge S, Placzek M, Blackshaw S | 2021 | Single-cell analysis of early chick hypothalamic development reveals that hypothalamic cells are induced from prethalamic-like progenitors | http://www.ncbi.nlm.nih.gov/geo/query/acc.cgi?acc=GSE171649 | NCBI Gene Expression Omnibus, GSE171649 |

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
