## [Editor Report]

The manuscript provides a comprehensive insight into the development of the tuberal hypothalamus of the chick by carefully analyzing the expression patterns of a plethora of proteins involved and perturbation of BMP signaling. The fundamental findings presented here substantially advance our understanding of the development of the chick tuberal hypothalamus from floor plate- like cells, mediated by an anterior to posterior wave of neuroepitelial-derived BMP signalling. Using bioinformatical and in situ profiling, fate mapping and tissue explants the authors present compelling evidence supporting their conclusion.

---

## [Decision Letter]

**Decision letter after peer review:**

[Editors’ note: the authors submitted for reconsideration following the decision after peer review. What follows is the decision letter after the first round of review.]

Thank you for submitting the paper "A neuroepithelial wave of BMP signalling drives anteroposterior specification of the tuberal hypothalamus" for consideration by *eLife*. Your article has been reviewed by 2 peer reviewers, one of whom is a member of our Board of Reviewing Editors, and the evaluation has been overseen by a Senior Editor. The reviewers have opted to remain anonymous.

Comments to the Authors:

We are sorry to say that, after consultation with the reviewers, we have decided that this work will not be considered further for publication by *eLife*.

This study integrates recent scRNA transcriptomics with high-resolution multiplexing in situ hybridization, fate mapping and tissue explants to unravel the spatiotemporal development of early chick tuberal hypothalamus. However, it does not convey the message well, and even for experts it is very hard to follow.

*Reviewer #1 (Recommendations for the authors):*

I recommend that each chapter conclude with a summary, which would help a reader not so familiar with the development of the chick tubular hypothalamus to evaluate the major findings.

The information presented in Materials and methods could be more detailed. For example, how many embryos were analyzed for each set of experiments? In the chapter on in-vivo manipulation of BMP signaling: I guess there should be more details on the Affi-Gel beads, at least a reference.

From my point of view, the discussion repeats too many details of the results. It would have been helpful if the authors could focus on the more general outcome of their results. E. g. they (i) could highlight the novelty of the findings, (ii) could describe how this process studied in chick differs from the corresponding process in mouse, and (iii) whether the way how this developmental process occurs has any similarity to other developmental processes, which result in a patterned tissue in a temporally and spatially distinct way. Or is it unique for chick development?

*Reviewer #2 (Recommendations for the authors):*

The paper is suitable for publication following further improvements in writing and figure arrangements to convey a clearer message.

1. The sequence of the figures (numerical and alphabetical) should follow the narrative context. For instance, on page 5, 'We therefore profiled the expression patterns of progenitor and neurogenic markers in detail, through multiplex hybridization chain reaction (HCR) in situ analyses of sagittal sections, transverse sections, wholemount ventral views and hemi-views (Supplementary Figure 1A-C).' Main figure should appear before supplementary figures.

2. The definitions of all the acronyms should be stated in both the text and legends if appear in the first time. E.g., figure 1K, no definition of all the acronyms; figure 1Q, confusing color coding with either gene or region name, please indicate both.

3. Some sentences are hard to understand, please consider rephrase. For example, page 6, 'at HH17, anterior tuberal progenitors are characterised by SIX6(high)-expression, and posterior tuberal progenitors, which overlie the tip of Rathke's pouch, are characterised by TBX2/SIX6(low) expression (Kim et al., 2022; Manning et al., 2006).'

4. Page 6, '…overlying the anterior-most SHH-expressing prechordal mesoderm (Figure 1, white arrow).' Should be Figure 1M and also some images in supplementary figure 1. Same applies for Figure 1N-P, U-W.

5. Lack of scale bar in figure 1R'-J'", supplementary figure 1 H'-J'" and boxed area…

6. Figure 2G, J, M, supplementary figure 3A, D please indicate the name of the color-coded region.

7. Page 12, 'These translocate to the HH17 region occupied by posterior tuberal progenitors (as judged through position and expression of FGF10/SIX6(low)), sometimes even just extending into the region occupied by anterior tuberal progenitors SIX6(high)/FGF10-negative (Figure 2O; Supplementary Figure 2D-H).' Confusing as the figures didn't show the mentioned genes, please indicate the color coded region or refer to correct figures.

8. Page 12-13, 'Triple label analyses show that cells maintain their relative spatial positions within the neuroectoderm between HH10 and HH17, with no evidence of mixing or dispersal (Figure 2M-O).' Hard to say unmixing without all time points and both green colored regions.

9. Page 13, last paragraph, please indicate the same gene/region name as in the figures, also please specify the genes/regions of the color code in each figure since the previous figure may not have the same time point. It is very confusing with multiple genes labeling same/different region of various time points and easily loss track.

10. Legends of Figure 2C, asterisk without explanation. Figure2D-F, consider change dot outline color, difference not obvious. Figure 2I, please indicate the acronyms of regions on the image. Figure 2L, no definition of the asterisk. Figure 2O, which one is the second injection? Please indicate which region it is injected to. Figure 2P, please indicate color coding here, it is not exactly the same as Figure 1Q.

11. Legend to figure S4A, 'In this embryo, TBX2 expression is not detected.' It looks like there are weak signals anterior.

. Legend Figure S5, 'Pink region in D depicts…, and yellow lines show the region dissected for explant culture.' What is the yellow dotted line region?

13. Page 24-25, 'Anterior explants cultured in control…did not express the supramammillary/floor plate marker, FOXA2, but…' In figure 5E', it seems FOXA2 is weakly expressed rather than none.

14. Page 25, 'Under these conditions, FOXA2 was lost and the explants expressed SIX6 and ISL1, but not TBX2 (Figure 5G-G"").' Figure 5G", TBX2 seems weakly expressed rather than none.

15. Signals quantifications of different genes in Figure 5E-H"", if possible?

16. Page 29. 'Ectopic BMPs significantly reduced, or eliminated, SIX6/SHH anterior tuberal progenitors…' Hard to understand this sentence, consider rephrasing.

17. Page 29, 'SHH expression was not appropriately downregulated…, and was marginally reduced in the region of the anterior ID…' 'No obvious changes in expression of the…FST, which at this stage…anterior ID'. Please explain what is 'anterior ID' in the text, not only in the legend.

18. Figure 6 and supplementary figure 7. I suggest to indicate the stages of the embryos by the side of the respective panels and in the legends.

19. Page 30 'Surprisingly, and in contrast to embryos examined at HH13,…or absent expression of TBX2 and FGF10…' The reduction of FGF10 is not obvious to me in supplementary figure 7F comparing to 7B.

20. Page 30, 'The loss of tuberal progenitor domains was accompanied by marked changes in anterior ventral forebrain morphology: the optic stalk failed to narrow, resulting in a gaping entrance to the optic stalk (Figure 6V) and copious SIX6/RAX/PAX6-positive eye tissue within (compare Figures 6F, H, with Figures 6L, N; individual channels shown in Figure 6T,T', U, U'; optic stalk entrance outlined in Figures 6R" versus 6S")…' The labeling sequence in this figure does not follow the text.

21. Page 30, '…the FOXA2 expression domain…appeared normal…' It seems to me that FOXA2 signal is increased in supplementary figure 7B and F, I and J.

22. Page 30, 'Expression of the telecephalic marker FOXG1,…to the optic stalk opening.' Please indicate the location of the optic stalk opening in the images.

23. Page 31, 'In control emrbyos, HES5, DLL1 and ELAVL4 are all detected in the hypothalamus…' Please indicate hypothalamus in the relevant images.

24. As a matter of style the Figure should not be mentioned in the discussion.

---

## [Author Response]

[Editors’ note: The authors appealed the original decision. What follows is the authors’ response to the first round of review.]

Reviewer #1 (Recommendations for the authors):I recommend that each chapter conclude with a summary, which would help a reader not so familiar with the development of the chick tubular hypothalamus to evaluate the major findings.

This is a good suggestion, and we now provide a summary at the end of each section

The information presented in Materials and methods could be more detailed. For example, how many embryos were analyzed for each set of experiments? In the chapter on in-vivo manipulation of BMP signaling: I guess there should be more details on the Affi-Gel beads, at least a reference.

We have revised the Materials and methods, to ensure all details are present. All n numbers are stated in the figure legends

From my point of view, the discussion repeats too many details of the results. It would have been helpful if the authors could focus on the more general outcome of their results. E. g. they (i) could highlight the novelty of the findings, (ii) could describe how this process studied in chick differs from the corresponding process in mouse, and (iii) whether the way how this developmental process occurs has any similarity to other developmental processes, which result in a patterned tissue in a temporally and spatially distinct way. Or is it unique for chick development?

This is a good suggestion, and we have now tightened the Discussion and focused on more general conclusions.

Reviewer #2 (Recommendations for the authors):The paper is suitable for publication following further improvements in writing and figure arrangements to convey a clearer message.1. The sequence of the figures (numerical and alphabetical) should follow the narrative context. For instance, on page 5, 'We therefore profiled the expression patterns of progenitor and neurogenic markers in detail, through multiplex hybridization chain reaction (HCR) in situ analyses of sagittal sections, transverse sections, wholemount ventral views and hemi-views (Supplementary Figure 1A-C).' Main figure should appear before supplementary figures.

We have rearranged several main Figures and associated Supplementary Figures. Major rearrangements have been made between Figure 1 and Figure 1—figure supplement 1; Figure 2 and Figure 2—figure supplement 1; Figure 2 and Figure 2—figure supplement 2; Figure 6 and Figure 6 supplement 1. Additionally, the previous panel Supplementary Figure 2B has been moved to Figure 7

The rearrangements ensure that the sequence of figures follows the narrative, and that the reader can follow critical conceptual points without having to jump from main to supplementary figures. All main figures now appear before supplementary figures.

2. The definitions of all the acronyms should be stated in both the text and legends if appear in the first time. E.g., figure 1K, no definition of all the acronyms; figure 1Q, confusing color coding with either gene or region name, please indicate both.

Acronyms are now stated in both text and legend.

We thank the reviewer for pointing out that our previous schematics – where we used colours to represent both regions and gene expression patterns – were confusing. We have altered the schematics, so that regions are now shown in colours, and gene expression profiles indicated by (non-coloured) bar lines (new schematics in Figure 1O-S, Figure 2Q).

3. Some sentences are hard to understand, please consider rephrase. For example, page 6, 'at HH17, anterior tuberal progenitors are characterised by SIX6(high)-expression, and posterior tuberal progenitors, which overlie the tip of Rathke's pouch, are characterised by TBX2/SIX6(low) expression (Kim et al., 2022; Manning et al., 2006).'

We have extensively re-written the paper, and have re-phrased this sentence

4. Page 6, '…overlying the anterior-most SHH-expressing prechordal mesoderm (Figure 1, white arrow).' Should be Figure 1M and also some images in supplementary figure 1. Same applies for Figure 1N-P, U-W.

We have extensively re-written this section, and simplified the figures.

5. Lack of scale bar in figure 1R'-J'", supplementary figure 1 H'-J'" and boxed area…

Thank you – we have addressed these issues

6. Figure 2G, J, M, supplementary figure 3A, D please indicate the name of the color-coded region.

As per point 2. We have altered the schematics, so that regions are now shown in colours, and gene expression profiles are indicated by (non-coloured) bar lines (new schematics in Figure 1O- S, Figure 2Q).

7. Page 12, 'These translocate to the HH17 region occupied by posterior tuberal progenitors (as judged through position and expression of FGF10/SIX6(low)), sometimes even just extending into the region occupied by anterior tuberal progenitors SIX6(high)/FGF10-negative (Figure 2O; Supplementary Figure 2D-H).' Confusing as the figures didn't show the mentioned genes, please indicate the color coded region or refer to correct figures.

This is a critical piece of data. The text did, in fact, refer to the correct figures, but the figures showed only a subset of the genes that were mentioned. We have therefore substantially revised Figure 2, and associated Figure 2—figure supplement 1, and show new data to fully support the narrative. New data is shown in Figure 2G-K’, Figure 2L-P’, Figure 2—figure supplement 2E-G. Data shown in previous Figure 2 has now been moved to Figure 2—figure supplement 2H-P.

8. Page 12-13, 'Triple label analyses show that cells maintain their relative spatial positions within the neuroectoderm between HH10 and HH17, with no evidence of mixing or dispersal (Figure 2M-O).' Hard to say unmixing without all time points and both green colored regions.

We now provide additional evidence to support this conclusion, showing new data where we have performed a time-lapse analysis of isolated neural tubes, after DiI/DiO labelling (new data shown in Figure 2—figure supplement 2Q-T). We agree, however, that we should state that there is ‘no apparent mixing’.

9. Page 13, last paragraph, please indicate the same gene/region name as in the figures, also please specify the genes/regions of the color code in each figure since the previous figure may not have the same time point. It is very confusing with multiple genes labeling same/different region of various time points and easily loss track.

As per point 2. We have altered the schematics, so that regions are now shown in colours, and gene expression profiles are indicated by (non-coloured) bar lines (new schematics in Figure 1O- S, Figure 2Q).

10. Legends of Figure 2C, asterisk without explanation. Figure2D-F, consider change dot outline color, difference not obvious. Figure 2I, please indicate the acronyms of regions on the image. Figure 2L, no definition of the asterisk. Figure 2O, which one is the second injection? Please indicate which region it is injected to. Figure 2P, please indicate color coding here, it is not exactly the same as Figure 1Q.

Figure 2 was a key piece of data- showing how particular HypFP cells fate-map to specific tuberal domains – as judged on the basis of both position and gene expression.

Given this comment (and comment 7 above) we have substantially revised Figure 2, and associated Figure 2—figure supplement 1, and show new data to fully support the narrative.

New data is shown in Figure 2G-K’, Figure 2L-P’, Figure 2—figure supplement 2E-G. Data shown in previous Figure 2 has now been moved to Figure 2—figure supplement 2H-P.

Additionally, we have addressed the specific comments:

Figure 2C-F: we have explained the asterisk, and altered the dotted outlines

Figures 2G-I and 2M-O have been moved to Figure 2-supplement figure 2H-P, and the regions more clearly explained through the new schematics.

11. Legend to figure S4A, 'In this embryo, TBX2 expression is not detected.' It looks like there are weak signals anterior.

The reviewer is correct – and we have altered the text. Note this does not affect any conclusions

12. Legend Figure S5, 'Pink region in D depicts…, and yellow lines show the region dissected for explant culture.' What is the yellow dotted line region?

We have replaced panel D to better illustrate the region dissected

13. Page 24-25, 'Anterior explants cultured in control…did not express the supramammillary/floor plate marker, FOXA2, but…' In figure 5E', it seems FOXA2 is weakly expressed rather than none.

In response to point 15, we now provide semi-quantitative analyses. FOXA2 is not detected in the majority of explants, but is detected at very low levels in a minority (new Figure 5F).

14. Page 25, 'Under these conditions, FOXA2 was lost and the explants expressed SIX6 and ISL1, but not TBX2 (Figure 5G-G"").' Figure 5G", TBX2 seems weakly expressed rather than none.

In response to point 15, we now provide semi-quantitative analyses. TBX2 is not detected in the majority of explants, but is detected at very low levels in a minority (new Figure 5J).

15. Signals quantifications of different genes in Figure 5E-H"", if possible?

Direct quantification after HRC analysis proved to be unreliable. We have therefore assigned explants to 3 distinct categories – those showing high expression in >50% explants; those showing weak expression in <50% explants, and those showing no expression. Representative examples are shown in Figure 5

16. Page 29. 'Ectopic BMPs significantly reduced, or eliminated, SIX6/SHH anterior tuberal progenitors…' Hard to understand this sentence, consider rephrasing.

We have re-written this section to clarify

17. Page 29, 'SHH expression was not appropriately downregulated…, and was marginally reduced in the region of the anterior ID…' 'No obvious changes in expression of the…FST, which at this stage…anterior ID'. Please explain what is 'anterior ID' in the text, not only in the legend.

We have explained the meaning of this acronym

18. Figure 6 and supplementary figure 7. I suggest to indicate the stages of the embryos by the side of the respective panels and in the legends.

This is a good suggestion, and we have indicated the stages in Figure 6. Note that we have rearranged some of the panels, moving the multiplex panels (previously in Figure 6) to Figure 6—figure supplement 1.

19. Page 30 'Surprisingly, and in contrast to embryos examined at HH13,…or absent expression of TBX2 and FGF10…' The reduction of FGF10 is not obvious to me in supplementary figure 7F comparing to 7B.

We thank the reviewer for pointing this out. This was signal bleed-through from the DAPI channel (in a previous round), which we have now reduced.

20. Page 30, 'The loss of tuberal progenitor domains was accompanied by marked changes in anterior ventral forebrain morphology: the optic stalk failed to narrow, resulting in a gaping entrance to the optic stalk (Figure 6V) and copious SIX6/RAX/PAX6-positive eye tissue within (compare Figures 6F, H, with Figures 6L, N; individual channels shown in Figure 6T,T', U, U'; optic stalk entrance outlined in Figures 6R" versus 6S")…' The labelling sequence in this figure does not follow the text.

We have rearranged the panels in Figures 6 and Figure 6-supplement figure 1 (previously named Supplementary Figure 7), so that they follow the text

21. Page 30, '…the FOXA2 expression domain…appeared normal…' It seems to me that FOXA2 signal is increased in supplementary figure 7B and F, I and J.

We do not think that FOXA2 expression is higher in the BMP-treated embryos but we have altered the wording to leave this as a possibility. Note this does not affect the main conclusions of the paper

22. Page 30, 'Expression of the telecephalic marker FOXG1,…to the optic stalk opening.' Please indicate the location of the optic stalk opening in the images.

We have now indicated the location of the optic stalk opening (Figure 6P, P’)

23. Page 31, 'In control emrbyos, HES5, DLL1 and ELAVL4 are all detected in the hypothalamus…' Please indicate hypothalamus in the relevant images.

We have outlined the hypothalamus in the relevant images (Figure 6T-V and T’-V’)

24. As a matter of style the Figure should not be mentioned in the discussion.

We have removed mention of Figures from the Discussion